# An essential *Staphylococcus aureus* cell division protein directly regulates FtsZ dynamics

Prahathees J Eswara[1,2]*, Robert S Brzozowski[2], Marissa G Viola[3], Gianni Graham[1,2], Catherine Spanoudis[2], Catherine Trebino[3], Jyoti Jha[4], Joseph I Aubee[5], Karl M Thompson[5], Jodi L Camberg[3,6], Kumaran S Ramamurthi[1]*

[1]Laboratory of Molecular Biology, National Cancer Institute, National Institutes of Health, Bethesda, United States; [2]Department of Cell Biology, Microbiology and Molecular Biology, University of South Florida, Tampa, United States; [3]Department of Cell and Molecular Biology, University of Rhode Island, Kingston, United States; [4]Laboratory of Biochemistry and Molecular Biology, National Cancer Institute, National Institutes of Health, Bethesda, United States; [5]Department of Microbiology, College of Medicine, Howard University, Washington, United States; [6]Department of Nutrition and Food Sciences, University of Rhode Island, Kingston, United States

**Abstract** Binary fission has been well studied in rod-shaped bacteria, but the mechanisms underlying cell division in spherical bacteria are poorly understood. Rod-shaped bacteria harbor regulatory proteins that place and remodel the division machinery during cytokinesis. In the spherical human pathogen *Staphylococcus aureus*, we found that the essential protein GpsB localizes to mid-cell during cell division and co-constricts with the division machinery. Depletion of GpsB arrested cell division and led to cell lysis, whereas overproduction of GpsB inhibited cell division and led to the formation of enlarged cells. We report that *S. aureus* GpsB, unlike other *Firmicutes* GpsB orthologs, directly interacts with the core divisome component FtsZ. GpsB bundles and organizes FtsZ filaments and also stimulates the GTPase activity of FtsZ. We propose that GpsB orchestrates the initial stabilization of the Z-ring at the onset of cell division and participates in the subsequent remodeling of the divisome during cytokinesis.
DOI: https://doi.org/10.7554/eLife.38856.001

**\*For correspondence:**
prahathees@usf.edu (PJE);
ramamurthiks@mail.nih.gov (KSR)

**Competing interests:** The authors declare that no competing interests exist.

## Introduction

Bacterial cell division has been extensively studied in rod-shaped organisms such as *Escherichia coli* and *Bacillus subtilis* (*Adams and Errington, 2009*; *Lutkenhaus et al., 2012*; *Rowlett and Margolin, 2015*; *Tsang and Bernhardt, 2015*). However, spherical bacteria lack several key components found in these well-studied model organisms (*Pinho et al., 2013*), so fundamental features of how they divide are poorly understood. The Gram-positive human pathogen *Staphylococcus aureus* is a spherical bacterium that is commensal in ~30% of the U.S. population (*Kuehnert et al., 2006*), but in immunocompromised individuals, it is a leading cause of bacteremia and nosocomial infections in industrialized nations (*Klevens et al., 2007*). The emergence of several antibiotic resistant strains of *S. aureus* has necessitated the identification of novel antibiotic targets (*Pendleton et al., 2013*). In recent years, components of the bacterial cell division machinery have been proposed as such targets (*Lock and Harry, 2008*; *Sass and Brötz-Oesterhelt, 2013*).

**eLife digest** A bacterium called *Staphylococcus aureus* causes many infections in humans, especially in hospital patients with weakened immune systems. These infections are generally treated with drugs known as antibiotics that interact with specific proteins in the bacteria to kill the cells, or stop them from growing. However, some *S. aureus* infections are resistant to the antibiotics currently available so there is a need to develop new drugs that target different bacterial proteins.

Bacteria multiply by dividing to make identical copies of themselves. When a bacterium is preparing to divide, filaments made of a protein called FtsZ form a ring at the site where the cell will split. Many other proteins are involved in controlling how and when a cell divides. For example, several species of bacteria harbor a dispensable cell division protein called GpsB. In at least one organism, it helps to maintain the proper shape of the cell during cell division. In *S. aureus*, though, GpsB is essential for cells to survive and could therefore be a potential target for new antibiotics. However, its role in *S. aureus* has not been studied.

Eswara et al. have now used genetic and biochemical approaches to study the *S. aureus* form of the GpsB protein. The experiments show that GpsB moves to the middle of *S. aureus* cells just before they begin to divide and binds directly to FtsZ. This helps to secure the position of FtsZ across the middle of the cell and activates the protein so that the cell can begin to divide into two. In cells that produce too much GpsB, the FtsZ proteins become active too early, leading to the cells growing larger and larger until they burst.

The findings of Eswara et al. reveal that GpsB plays a different role in *S. aureus* cells than in some other species of bacteria. Further studies into such differences could help researchers to develop new antibiotics, as well as improving our understanding of why bacteria are so diverse.

DOI: https://doi.org/10.7554/eLife.38856.002

GpsB is a small coiled-coil cell division protein (*Claessen et al., 2008*; *Rismondo et al., 2016*; *Tavares et al., 2008*) that is widely conserved in the Firmicutes phylum and is conditionally required for growth in certain species, depending on growth media and temperatures (*Claessen et al., 2008*; *Fleurie et al., 2014*; *Land et al., 2013*; *Rismondo et al., 2016*; *Tavares et al., 2008*). GpsB is highly co-conserved (*Pinho et al., 2013*) with the cell division protein DivIVA. Like DivIVA, GpsB is relatively small and harbors a highly homologous N-terminal α-helical domain. However, the C-terminus differs from that of DivIVA: whereas DivIVA assembles into an anti-parallel tetramer, the GpsB structure was reported to hexamerize with a parallel alignment of helices (*Rismondo et al., 2016*). Similar to DivIVA (*Kaval and Halbedel, 2012*), GpsB orthologs perform slightly different functions in different species. In the rod-shaped *Bacillus subtilis* and *Listeria monocytogenes*, GpsB participates in shuttling a cell wall assembly protein (PBP1 or PBP A1, respectively) to help maintain the characteristic rod shape of the bacterium (*Claessen et al., 2008*; *Rismondo et al., 2016*). In the ovoid-shaped *Streptococcus pneumoniae*, GpsB additionally has been reported to interact with PBP2a, PBP2b, and MreC, and has been implicated in recruiting a Ser/Thr kinase to mid-cell that activates cell wall assembly machinery specifically at the division septum, thereby modulating a switch between peripheral and medial cell wall assembly to again maintain the proper shape of the cell (*Fleurie et al., 2014*; *Rued et al., 2017*). In all reported cases, GpsB interacts with a peripheral divisome component, EzrA, but not necessarily core components of the division machinery, to mediate its role in cell shape maintenance (*Claessen et al., 2008*; *Fleurie et al., 2014*; *Rued et al., 2017*; *Steele et al., 2011*).

In *S. aureus*, GpsB is an essential protein (*Santiago et al., 2015*) (M. Santiago, personal communication), but its cellular function is poorly understood. Herein, we report that GpsB interacts directly with bacterial tubulin homolog FtsZ, the core component of the division machinery, and orchestrates the dynamics of its assembly. In vivo, we show that GpsB localizes to mid-cell at the onset of cell division and co-constricts with the divisome during cytokinesis. Depletion of GpsB in vivo arrested cell division and prevented the robust assembly of the divisome at mid-cell. In vitro, we show that purified GpsB promotes lateral interactions between FtsZ polymers in a manner reminiscent of bundling, thereby increasing the local concentration of FtsZ, and organizes the polymers. Unlike other proteins that exhibit FtsZ bundling activity, GpsB stimulated FtsZ GTPase activity. Consistent with

this activity, overproduction of GpsB in vivo inhibited cell division and resulted in the production of large *S. aureus* cells. Our data suggest that, compared to GpsB orthologs in other Gram-positive bacteria, *S. aureus* GpsB plays a significantly different role by directly interacting with central component of the division machinery to regulate the remodeling of the divisome during cytokinesis: first, by bundling and stabilizing FtsZ polymers at mid-cell by promoting lateral interactions between FtsZ filaments, which increases the local concentration and triggers the GTPase activity of FtsZ and allows cytokinesis to proceed.

## Results

### Overproduction of *S. aureus* GpsB inhibits cell division in *B. subtilis* and *S. aureus*

To initially investigate if *Staphylococcal* GpsB (GpsB[Sa]) performs a similar function as the *B. subtilis* GpsB ortholog (GpsB[Bs]), we expressed *gpsB*[Sa] under the control of an inducible promoter in *B. subtilis*. In the presence of inducer, otherwise WT *B. subtilis* harboring either *gpsB*[Sa] or *gpsB*[Sa]-GFP exhibited a severe growth defect (*Figure 1*). In contrast, cells similarly expressing *gpsB*[Bs] or *gpsB*[Bs]-GFP did not exhibit a growth defect (*Figure 1A*), suggesting that cell toxicity was specifically due to expression of the *S. aureus* ortholog of *gpsB* (*Figure 1—figure supplement 1A*). Immunoblotting with antisera specific to GpsB[Sa] revealed a ~ 3.2 fold overproduction of GpsB[Sa] at a population level in the presence of inducer (*Figure 1—figure supplement 1B*; note that the anti-GpsB[Sa] antiserum did not recognize GpsB[Bs]). In the absence of inducer, *B. subtilis* cells harboring *gpsB*[Sa] examined by epifluorescence microscopy were of uniform length and displayed division septa at mid-cell (*Figure 1B*), but in the presence of inducer, these cells were filamentous with segregated chromosomes that rarely elaborated division septa (*Figure 1C*). GpsB interacts with several cell division proteins in different Gram-positive bacteria (*Claessen et al., 2008*; *Cleverley et al., 2016*; *Pompeo et al., 2015*). Deletion of *ezrA*, *ponA*, *prkC*, or *gpsB* resulted in minor morphological defects in *B. subtilis*, but overproduction of GpsB[Sa] in these strain backgrounds nonetheless resulted in filamentation (*Figure 1D–K*). Additionally, while deletion of *divIVA* resulted in cell elongation (*Edwards and Errington, 1997*), overproduction of GpsB[Sa] in the absence of DivIVA resulted in further filamentation (*Figure 1L–M*). Thus, the *B. subtilis* filamentation phenotype resulting from GpsB[Sa] overproduction does not require these peripheral cell division factors. We next examined if GpsB[Sa] affects FtsZ localization. In the absence of inducer, FtsZ-GFP localized properly to mid-cell at incipient and active sites of cell division (*Figure 1N*). However, upon overproduction of GpsB[Sa], filamentous cells displayed diffuse localization of FtsZ-GFP in the cytosol (*Figure 1O*), suggesting that GpsB[Sa] overproduction interferes with the localization of the central component of the *B. subtlis* cell division machinery.

In *B. subtilis*, *gpsB* is not essential for growth, but deletion of *ezrA* (a peripheral component of the divisome) together with *gpsB* is synthetically lethal (*Claessen et al., 2008*). In *B. subtilis* cells harboring a *gpsB* deletion and expressing *gpsB*[Bs], we obtained 865 ± 157 transformants harboring a deletion in *ezrA* (n = 3, per ~400 ng of transformed DNA containing *ezrA* deletion), whereas we did not recover any transformants when we attempted to delete *ezrA* in Δ*gpsB* cells that expressed *gpsB*[Sa], indicating that *gpsB*[Sa] was unable to complement the *gpsB*[Bs] deletion phenotype. Together with the different phenotypes observed upon overexpression of either *gpsB*[Bs] or *gpsB*[Sa] in *B. subtilis*, the data suggested that *S. aureus* GpsB may exhibit a different function or activity.

To test the effect of GpsB[Sa] overproduction in *Staphylococci*, we cloned *gpsB*[Sa] in a high copy plasmid under control of an inducible promoter, introduced the construct into *S. aureus* strain SH1000, stained the cells with a fluorescent membrane dye, and examined cell size using epifluorescence microscopy. Immunoblotting revealed an ~5.4 fold overproduction of GpsB at a population level relative to endogenous levels of GpsB (*Figure 1—figure supplement 1C*). 100% of WT cells we observed (n = 676) were less than 1.2 μm in diameter, as were WT cells harboring the empty vector (n = 100) (*Figure 2A–D*). In the absence of inducer, 6.4% (n = 971) of cells harboring the inducible copy of *gpsB*[Sa] were larger than 1.2 μm in diameter; in the presence of inducer, 56.9% (n = 770) of cells were larger than 1.2 μm (*Figure 2E–F*). Interestingly, overproduction of *gpsB*[Bs] did not result in a similar enlargement of *S. aureus* cells (*Figure 2G–H*), suggesting that the cell division inhibition phenotype in *B. subtilis* and *S. aureus* was unique to the overproduction of the *S. aureus* ortholog of

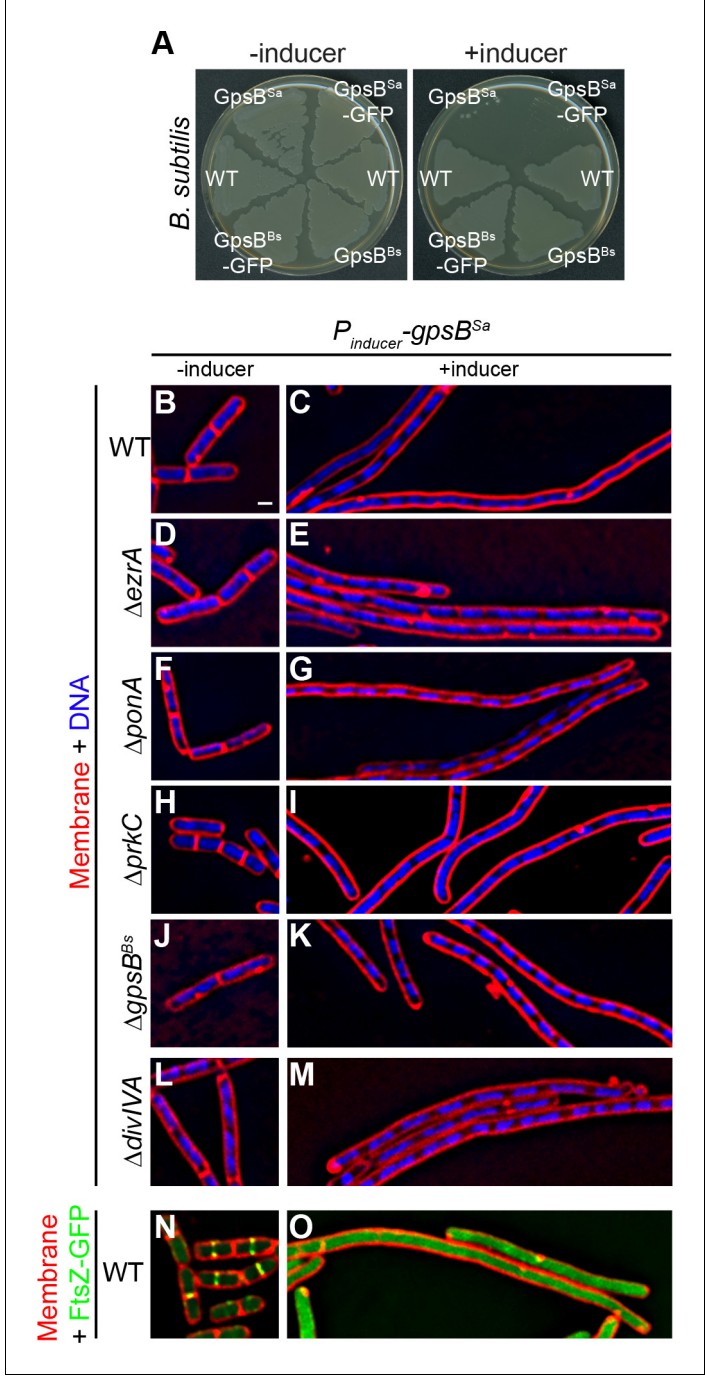

**Figure 1.** Overexpression of *S.aureus gpsB* inhibits cell division in *B. subtilis*. (**A**) Luria-Bertani agar plates streaked with wild type *B. subtilis* (WT, strain PY79), or otherwise wild type *B. subtilis* harboring an inducible copy of *gpsB^Bs* (GG18), *gpsB^Bs*-*gfp* (GG19), *gpsB^Sa* (GG7), or *gpsB^Sa*-*gfp* (GG8) integrated into the chromosome, in the absence (left) or presence (right) of inducer. (**B–M**) Morphology of cells of different deletion mutants of *B. subtilis* (Δ*ezrA*, GG35; Δ*ponA*, CS26; Δ*prkC*, CS24; Δ*gpsB*, CS40; Δ*divIVA*, CS94) harboring an inducible copy of *gpsB^Sa* grown in the absence (**B, D, F; H, J, L**) or presence (**C, E, G, I, K, M**) of inducer. (**N–O**) Localization of FtsZ-GFP in a strain (GG9) harboring an inducible copy of *gpsB^Sa* grown in the absence (**N**) or presence (**O**) of inducer. Membranes (red; **B–O**) visualized using the fluorescent dye FM4-64; chromosomes (blue; **B–M**) visualized using DAPI; FtsZ-GFP localization (green; **N–O**). Scale bar: 1 μm. Genotypes are listed in Key Resources Table.

DOI: https://doi.org/10.7554/eLife.38856.003

The following figure supplements are available for figure 1:

*Figure 1 continued on next page*

*Figure 1 continued*

**Figure supplement 1.** GpsB sequence and subcellular distribution in *S.aureus*.
DOI: https://doi.org/10.7554/eLife.38856.004

**Figure supplement 2.** GpsB[Sa]-GFP localizes to division septa in *B.subtilis*.
DOI: https://doi.org/10.7554/eLife.38856.005

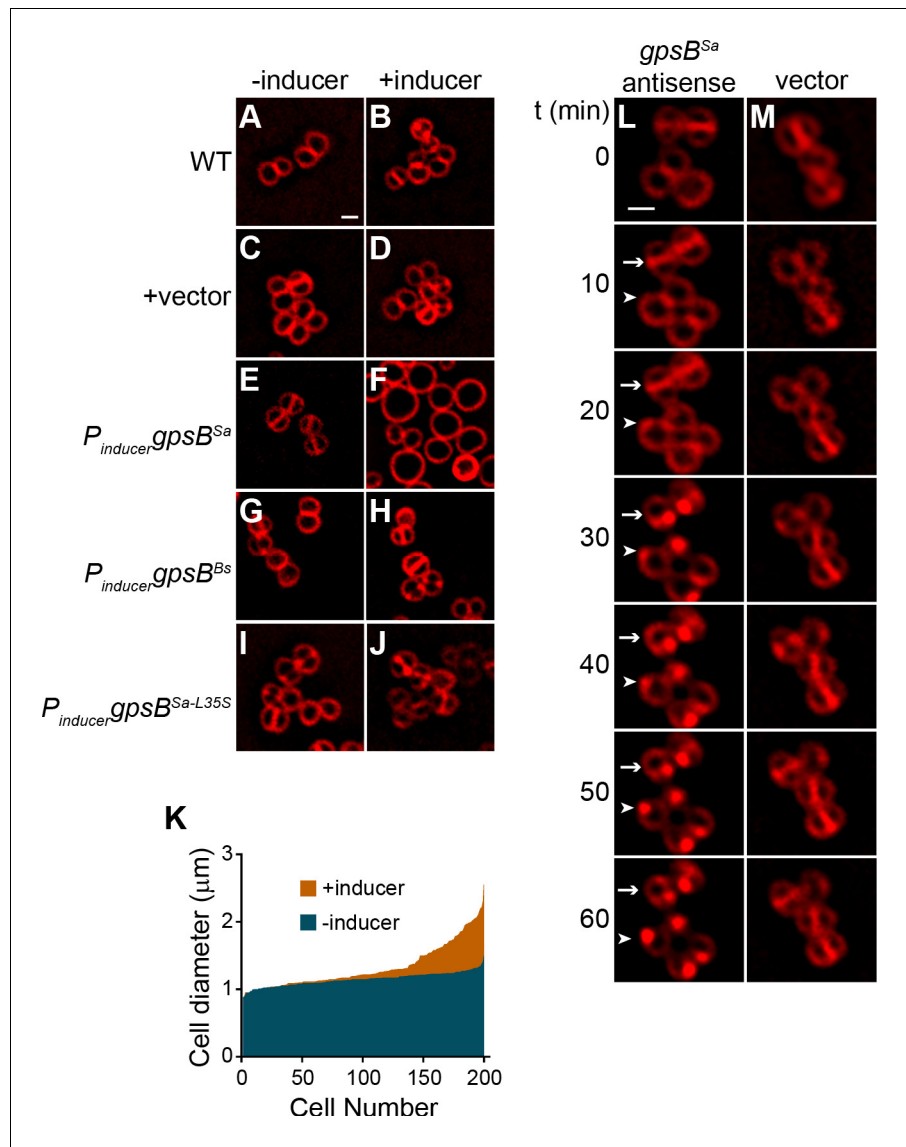

**Figure 2.** Overexpression and depletion phenotypes of *gpsB* in *S.aureus*. (**A–J**) Morphology of wild type *S. aureus* cells (A-B; strain SH1000) or *S. aureus* strains harboring a plasmid encoding an inducible copy of *gpsB*[Sa] (**E–F**; plasmid pPE45), vector backbone alone (**C–D**; pCL15), *gpsB*[Bs] (**G–H**; pPE83), or *gpsB*[Sa-L35S] (**I–J**; pPE79) in the absence (**A, C, E, G, I**) or presence (**B, D, F, H, J**) of inducer. (**K**) Histogram of cell diameters of 200 individual *S. aureus* cells harboring an inducible copy of *gpsB*[Sa] grown in the absence (blue) or presence (orange) of inducer IPTG. (**L–M**) Morphology of *S. aureus* cells harboring a plasmid encoding an inducible copy of antisense RNA against *gpsB* (L; SH1000 pGG59) or empty vector (**M**) at times (min) indicated to the left after induction. Arrow indicates a cell that has already initiated cell division when GpsB depletion was initiated; arrowhead indicates a cell that had not initiated cell division at the time of GpsB depletion. Membranes visualized using FM4-64. Scale bar: 1 µm.

DOI: https://doi.org/10.7554/eLife.38856.006

GpsB. Quantification of cell diameters of 200 individual cells overproducing GpsB[Sa] revealed a range of cell diameters higher than 1.2 µm in over half of the cells (*Figure 2K*). The variation in cell diameters was likely due to unequal expression of *gpsB[Sa]* in every cell, since control experiments in which *gfp* was placed under control of the inducible promoter revealed that only ~34% of cells (n = 263) produced GFP in the presence of inducer.

The *gpsB* gene is essential for viability in *S. aureus* (*Santiago et al., 2015*). Consistent with this observation, we were unable to knockout the gene, even in the presence of a complementing multi-copy plasmid, presumably due to the disruptive overproduction phenotype described above. We therefore sought to deplete GpsB by overexpressing *gpsB* antisense RNA under the control of an inducible promoter from a multicopy plasmid and examined the morphology of cells using fluorescence microscopy (antisense resulted in ~2.5-fold reduction in GpsB; *Figure 1—figure supplement 1F*). Immediately after addition of the inducer, cells harboring this construct were morphologically similar to cells harboring the empty vector (*Figure 2L–M*). At later time points, we observed that cells harboring the depletion construct that had already elaborated a division septum (*Figure 2L*, arrow) did not complete cytokinesis. Instead, the division septa became deformed and membrane aberrantly accumulated as foci. Cells that had not yet initiated cell division at the time of induction (*Figure 2L*, arrowhead) did not elaborate division septa and also accumulated aberrant membrane foci. In contrast, cells harboring only the empty vector (*Figure 2M*) elaborated division septa and completed cytokinesis during the observation period.

The severe growth defect imposed by *gpsB[Sa]* overexpression in *B. subtilis* permitted us to isolate suppressor mutations that could correct this defect. One such mutation, an intragenic single nucleotide change in *gpsB[Sa]*, altered the specificity of a highly conserved codon at position 35 from Leu to Ser (*Figure 1—figure supplement 1A*, boxed residue), and allowed *B. subtilis* cells overexpressing *gpsB[Sa-L35S]* to grow normally. To check if the L35S substitution caused in a major structural change in the protein, we purified WT GpsB[Sa] and the L35S variant and examined the α-helical content of both proteins using circular dichroism (CD) spectroscopy (*Figure 1—figure supplement 1D*). The CD spectrum revealed similar profiles for each protein, suggesting that the L35S substitution did not grossly affect the secondary structure of GpsB[Sa]. In the presence of inducer, *S. aureus* cells harboring inducible *gpsB[Sa-L35S]* did not exhibit a cell enlargement defect (*Figure 2I–J*). Taken together, we conclude that overproduction of GpsB[Sa], but not GpsB[Bs], inhibits cell division in both *S. aureus* and *B. subtilis*, resulting in cell filamentation (in *B. subtilis*) or cell enlargement (in *S. aureus*), like the depletion phenotype of FtsZ (*Pinho and Errington, 2003*). Depletion of GpsB in *S. aureus*, however, arrested cell division without a coincident enlargement of cells and ultimately caused aberrant membrane accumulation. Furthermore, substitution of Leu35 to Ser abolished the toxicity resulting from GpsB overproduction, suggesting that this residue is critical for GpsB[Sa] function.

## GpsB dynamically localizes to mid-cell in *S. aureus* and co-constricts with the division septum

We next examined the subcellular localization of GpsB[Sa]-GFP in *S. aureus*. In non-dividing cells GpsB[Sa]-GFP (produced at lower levels that did not result in cell division inhibition) localized near the cell periphery (*Figure 3A*, arrowhead). In dividing cells, GpsB[Sa]-GFP localized to mid-cell, between the segregated chromosomes, and co-localized with the constricting membrane (*Figure 3A*, arrow). In contrast, GpsB[Sa-L35S]-GFP localized primarily in the cytosol in both dividing and non-dividing cells (*Figure 3A*). Likewise, when produced at lower levels in *B. subtilis*, GpsB[Sa]-GFP accumulated at division septa, whereas the L35S variant localized primarily in the cytosol (*Figure 1—figure supplement 2*). Since the *L. monocytogenes* GpsB ortholog is membrane-associated (*Rismondo et al., 2016*), we next tested if the L35S substitution could have disrupted any intrinsic membrane affinity of GpsB[Sa] by fractionating *S. aureus* cell extracts and examining the distribution of GpsB[Sa]-GFP and GpsB[Sa-L35S]-GFP by immunoblotting (*Figure 1—figure supplement 1E*). Unlike *L. monocytogenes* GpsB, we detected *S. aureus* GpsB-GFP exclusively in the soluble fraction, suggesting that it does not directly associate with the *Staphylococcal* membrane. GpsB[Sa-L35S]-GFP was similarly detected in the cytosolic fraction. Association of *L. monocytogenes* GpsB with the membrane is reportedly mediated by Leu24, since substitution of Leu24 with Ala disrupted membrane association (*Rismondo et al., 2016*). Interestingly, the corresponding residue in *S. aureus* GpsB is Ala (*Figure 1—figure supplement 1A*, asterisk), consistent with the apparent lack of intrinsic affinity of *S. aureus* GpsB for the membrane. We conclude that, unlike *L. monocytogenes* GpsB, *S. aureus* GpsB (hereafter, simply

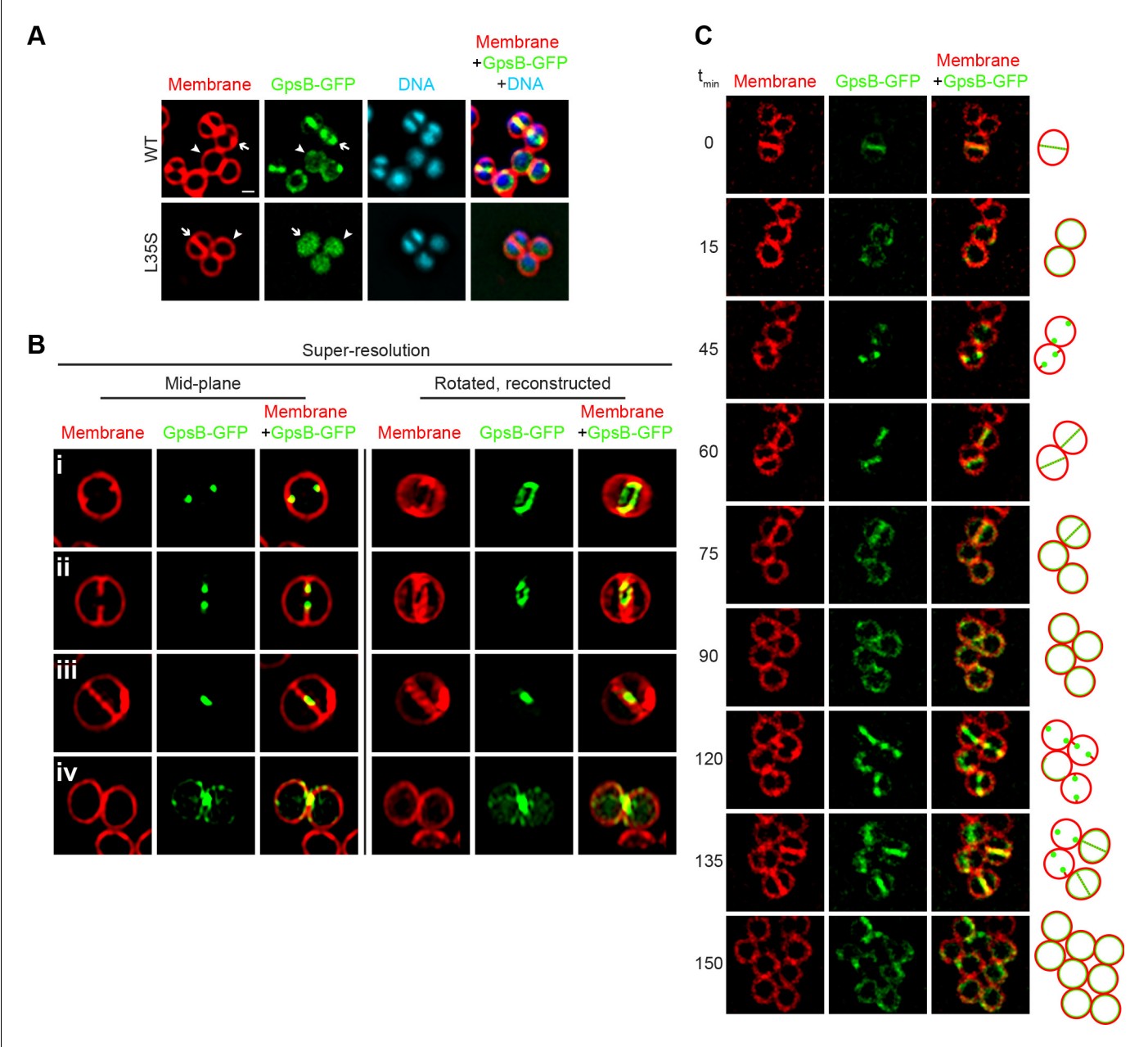

**Figure 3.** Dynamic redistribution of GpsB to mid-cell and periphery of *S.aureus* during the cell cycle. (**A**) Localization of GpsB-GFP (top, SH1000 pPE46) or GpsB[L35S]-GFP (bottom, SH1000 pPE80) to mid-cell in actively dividing cells (arrow) and to the periphery of cells that are not dividing (arrowhead). First panel: membranes visualized using FM4-64; second panel: GFP fluorescence; third panel: chromosomes visualized using DAPI; fourth panel: overlay of membrane, GFP, and DNA. (**B**) GpsB-GFP localization in *S. aureus* cells at various stages of division (**i–iv**) using structured illumination microscopy (SIM). First column: membranes visualized using FM4-64; second column: GpsB-GFP fluorescence; third column: overlay, membrane and GpsB-GFP. Columns 4–6: reconstruction of deconvolved Z-stacks and rotation of the cells in columns 1–3, respectively, around the vertical axis. (**C**) Time-lapse fluorescence micrographs of a dividing *S. aureus* cell taken at the time intervals indicated at the left. Left panels: membranes visualized using FM4-64; middle panels: GpsB-GFP fluorescence; right panels: overlay, membrane and GpsB-GFP. Depictions of GpsB-GFP localization patterns are to the right of the panels. Scale bar: 1 μm.

DOI: https://doi.org/10.7554/eLife.38856.007

'GpsB') is likely not directly membrane-associated and that cell peripheral localization of *S. aureus* GpsB may be mediated by another factor.

To discern if GpsB-GFP co-localized with, or at sites adjacent to, the site of membrane constriction, we employed structured illumination microscopy (SIM) (*Gustafsson, 2005*), a super-resolution technique that previously provided enough resolution to discern the localization of DivIVA-GFP on either side of the ~80 nm division septum (*Eswaramoorthy et al., 2011*). At the onset of cell division, mid-plane images of *S. aureus* cells displayed only two GpsB-GFP foci that co-localized with sites of membrane invagination at mid-cell (*Figure 3Bi*). Reconstruction of deconvolved Z-stacks and rotation of the reconstructed image around the axis of cell division revealed that GpsB-GFP formed an irregular ring-shaped structure, reminiscent of the structure of an assembling divisome (*Figure 3Bi*) (*Lund et al., 2018*). In cells that were further advanced in cell division, the two foci of GpsB-GFP followed the leading edges of the constricting membrane (*Figure 3Bii*) and formed a ring structure that was smaller than the diameter of the cell, (*Figure 3Bii*) (*Buss et al., 2015*; *Ebersbach et al., 2008*), suggesting that the GpsB ring structure co-constricts with the division machinery. In a cell approaching completion of cytokinesis, GpsB-GFP collapsed into a single focus at the center of the invaginating membrane (*Figure 3Biii*), and immediately after the completion of cell division, we observed that GpsB-GFP localized largely to the cell periphery in the adjacent daughter cells (*Figure 3Biv*), suggesting that GpsB may dynamically localize during the cell cycle.

Phototoxicity induced by SIM precluded us from performing super-resolution time lapse experiments of actively dividing cells using this method. To test the dynamic nature of GpsB-GFP localization, we followed the fate of GpsB-GFP in individual cells through three rounds of cell division using diffraction-limited epifluorescence microscopy. At the onset of our measurements, GpsB-GFP localized primarily at mid-cell in a cell that had completed cytokinesis and was poised to separate into two daughter cells (*Figure 3*; *[Steele et al., 2011]*). After cell separation, GpsB-GFP redistributed to the periphery of each daughter cell (*Figure 3*). Beginning the next round of cell division, GpsB-GFP re-localized to the mid-cell of each daughter cell as two foci that coincided with the invaginating membrane (*Figure 3C,t*). It again localized with the invaginating membrane, followed by redistribution of fluorescence to the cell peripheries of the daughter cells (*Figure 3*). The redistribution of peripherally-localized GpsB to the division septum is reminiscent of the FtsZ-dependent late localization of GpsB reported in *S. pneumoniae* (*Land et al., 2013*). We therefore conclude that GpsB localizes as a single ring-shaped structure at mid-cell at the onset of cell division, constricts with the invaginating membrane during cytokinesis, and ultimately, after daughter cell separation, uniformly redistributes to the periphery of each daughter cell.

## GpsB localization and divisome assembly reciprocally influence each other

Although the *S. aureus* ortholog of GpsB was non-functional in *B. subtilis*, its ability to localize at mid-cell suggested that it is capable of recognizing an intrinsic feature of the divisome shared between *B. subtilis* and *S. aureus* (*Figure 1—figure supplement 2*). The bacterial divisome is composed of approximately 10 core proteins (*Lutkenhaus et al., 2012*; *Margolin, 2005*) that direct the cell wall assembly machinery to mid-cell and mediate cell membrane constriction during cell division. The core divisome component is the bacterial tubulin homolog, FtsZ (*Coltharp et al., 2016*; *Osawa and Erickson, 2013*), which is a target of cell division regulators in different systems (*Ortiz et al., 2016*). To investigate a potential interaction between the divisome and GpsB, we examined the localization of GpsB-GFP in *S. aureus* cells grown in the presence and absence of the PC190723, a small ligand that inhibits GTPase activity of FtsZ and inhibits cell division (*Andreu et al., 2010*; *Haydon et al., 2008*). In the presence of the drug, 92.5% of cells (n = 200) harboring empty vector exhibited a diameter larger than 1.5 µm, compared to just 30.5% of cells in the absence of inhibitor, consistent with a block in cell division (*Figure 4A–B'*). To confirm that the drug was inhibiting divisome assembly, we examined the localization of ZapA-GFP, a known early stage cell division protein that assembles concomitantly with FtsZ (*Gamba et al., 2009*; *Reichmann et al., 2014*) and is used as a proxy for localization of FtsZ. In untreated cells, ZapA localized at mid-cell at the onset of cell division (*Figure 4C–C'*), but in the presence of the inhibitor, 96.5% of cells (n = 200) displayed diffuse and/or punctate localization in the cytosol that was not located at mid-cell (*Figure 4D–D'*), indicating that the divisome was not assembling correctly due to inhibition of FtsZ. In the absence of inhibitor, GpsB-GFP localized at mid-cell or the periphery in 55%

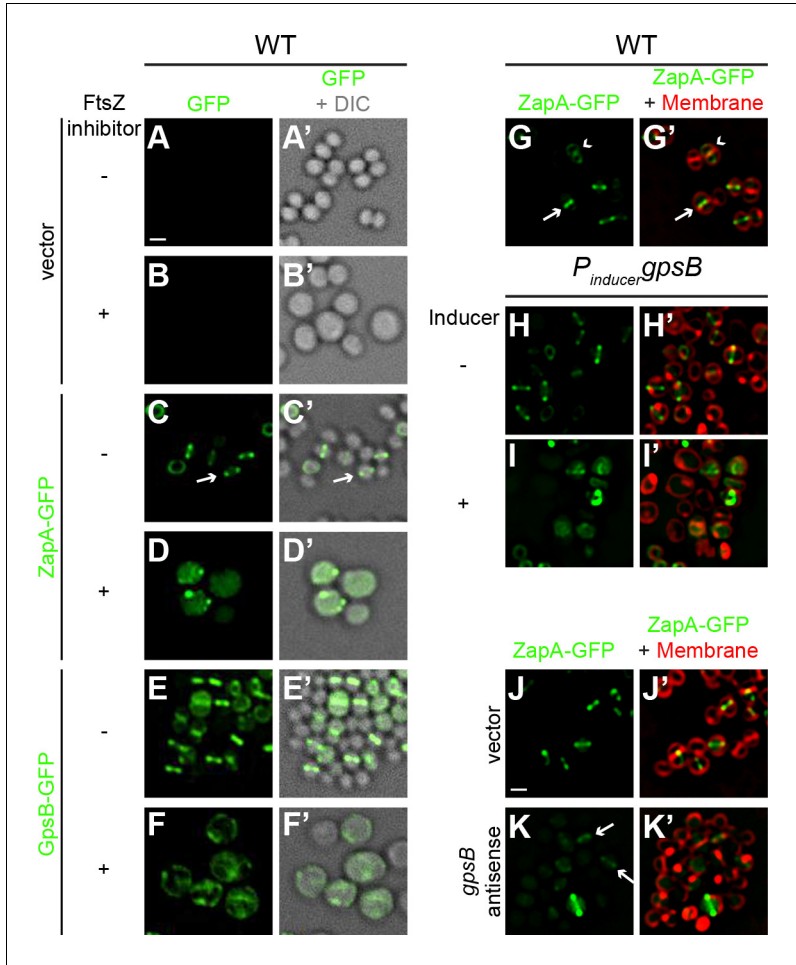

**Figure 4.** FtsZ and GpsB reciprocally influence each other's subcellular localization pattern. Morphology of wild type *S. aureus* cells harboring empty vector (strain SH1000 pCL15) in the absence (**A–A'**) or presence (**B–B'**) of FtsZ inhibitor PC190723 as visualized in the GFP fluorescence channel (**A–B**) or by differential interference contrast (DIC; **A'–B'**). Localization of ZapA-GFP (proxy for FtsZ localization; strain SH1000 pRB42; **C–D'**) or GpsB-GFP (**E-F'**; SH1000 pPE46) in the absence (**C-C'; E-E'**) or presence (**D-D'; F-F'**) of *S. aureus* FtsZ GTPase activity inhibitor PC190723. (**G–I'**) Localization of ZapA-GFP in wild type (**G-G'**; SH1000 pRB42), or in cells harboring an IPTG-inducible copy of *gpsB* in the absence (**H-H'**; SH1000 pPE45 pRB42) or presence (**I–I'**) of IPTG. Localization of ZapA-GFP in cells harboring vector only (**J-J'**; SH1000 pRB42 pEPSA5) or vector producing *gpsB* antisense RNA (**K-K'**; SH1000 pRB42 pGG59). A-K: GFP fluorescence; **A'-F'**: overlay of GFP fluorescence and DIC; **G'-K'**: overlay of GFP fluorescence and membrane. Arrows indicate site of cell division; arrowheads indicate ZapA-GFP localization at the subsequent plane of cell division. Scale bar: 1 µm.

DOI: https://doi.org/10.7554/eLife.38856.008

or 30% of cells, respectively, and mis-localized in the cytosol in 15% of cells (n = 200; *Figure 4E–E'*). In contrast, in the presence of inhibitor, GpsB did not localize to mid-cell in any cell observed (n = 200) and instead displayed a combination of diffuse cytosolic localization and aggregation along the cell periphery (*Figure 4F–F'*). The data therefore indicated that that GpsB localization to mid-cell depends directly or indirectly on functional FtsZ.

We next tested how GpsB influences divisome assembly. In otherwise wild type cells producing ZapA-GFP, no cell enlargement was detected; among them, 53% of cells displayed ZapA-GFP localized to mid-cell (these were cells that were actively undergoing cell division (*Figure 4G–G'*, arrow) and 37% of cells displayed ZapA-GFP as a ring that corresponded to the subsequent plane of cell division in daughter cells that had recently completed cytokinesis (*Figure 4G–G'*, arrowhead; n = 200). In contrast, in cells harboring inducible *gpsB*, addition of inducer resulted in the

enlargement of cells and ZapA-GFP was mis-localized in 86% of the enlarged cells (n = 100; *Figure 4H–I′*). Assuming that the FtsZ bundling activity of ZapA is not synergistically participating with GpsB overexpression, this suggests that the cell enlargement phenotype caused by overproduction of GpsB was due to the mis-assembly of the divisome.

To determine the behavior of the divisome in GpsB-depleted cells, we monitored the localization of ZapA-GFP. In cells harboring empty vector, ZapA-GFP localized to mid-cell in 82.5% of the cells (n = 200; *Figure 4J–J′*). Quantification of fluorescence intensity in individual cells revealed that the fluorescence of mid-cell-localized ZapA-GFP was 2429 ± 1346 units/cell (n = 75). At earlier time points after induction to deplete GpsB, before cell lysis, we observed faint ZapA-GFP signals at mid-cell in 41% of the cells and diffuse localization in the remaining cells (n = 200; *Figure 4K–K′*), but the mean fluorescence intensity of the ZapA-GFP ring structure (614 ± 450 units/cell; n = 75) was nearly four-fold lower than that observed for ZapA-GFP intensity in the absence of *gpsB* depletion. Together, the observations suggest that divisome assembly and GpsB localization are reciprocally influenced: GpsB requires FtsZ for localization to mid-cell; overproduction of GpsB disrupts divisome assembly; and depletion of GpsB prevents robust divisome assembly at mid-cell that precedes membrane deformities that ultimately lead to cell lysis.

## GpsB stimulates GTPase activity of FtsZ

To test if GpsB directly influences FtsZ behavior, we purified recombinant *S. aureus* FtsZ, GpsB, and GpsB[L35S] (*Figure 5A*) and examined the GpsB variants by size exclusion chromatography (*Figure 5B*). GpsB eluted in two peaks by size exclusion chromatography, which approximately corresponds to the predicted sizes of hexameric (*Rismondo et al., 2016*) and dodecameric GpsB (*Figure 5B*, top), indicating that *S. aureus* GpsB could potentially exist in two forms. In contrast, GpsB[L35S] eluted exclusively as a dodecamer (*Figure 5B*, bottom), suggesting that its inability to form hexamers could underlie its loss of function in vivo.

We next measured the GTP hydrolysis activity of purified *S. aureus* FtsZ with time at increasing protein concentrations. Unlike the well-characterized *E. coli* FtsZ, which robustly hydrolyzes GTP (*Arjes et al., 2015*; *Buske and Levin, 2012*; *Mukherjee and Lutkenhaus, 1998*; *Romberg and Levin, 2003*), *S. aureus* FtsZ hydrolyzed GTP poorly below ~30 μM (*Figure 5C*) (*Anderson et al., 2012*). The rate of hydrolysis continued to increase with FtsZ concentration (*Figure 5D*), displaying a behavior more similar to FtsZ from the Gram-positive *Streptococcus pneumoniae*, which has a critical concentration above 10 μM, than to *E. coli* FtsZ (although a lag observed for *S. pneumonia* FtsZ was not detected for *S. aureus* FtsZ) (*Salvarelli et al., 2015*). It should be noted that this result contrasts with that of Elsen et al., which reported a low critical concentration for *S. aureus* FtsZ (~5 μM) (*Elsen et al., 2012*). However, a recent report by Wagstaff et al. showed GTP hydrolysis rates at high *S. aureus* FtsZ concentration (10 and 20 μM), and similar to the rates reported here (*Wagstaff et al., 2017*). These differences could be due to varying populations of conformationally active FtsZ in different preparations (*Elsen et al., 2012*).

We next measured the effect of GpsB on the GTP hydrolysis rate of FtsZ. Incubation of 30 μM FtsZ with increasing amounts of GpsB resulted in a non-linear stimulation of GTP hydrolysis activity, wherein appreciable stimulation of GTP hydrolysis was only seen above 8 μM GpsB (*Figure 5E*). At 10 μM GpsB (1:3 ratio of monomeric GpsB:FtsZ; 1:18 ratio of hexameric GpsB:FtsZ; 1:36 ratio of dodecameric GpsB:FtsZ), GTP hydrolysis was stimulated ~3 fold. As a control, GpsB alone did not exhibit appreciable GTPase activity (*Figure 5E*). In contrast, incubation of FtsZ with GpsB[L35S] did not appreciably stimulate GTPase activity of FtsZ (*Figure 5F*), nor did the addition of GpsB, even at equimolar amounts, to lower concentrations (10 μM) of FtsZ. Thus, wild type GpsB, which purifies as a hexamer and dodecamer, stimulates the GTPase activity of FtsZ at substoichiometric levels at sufficiently high enough concentrations of FtsZ (above 30 μM), whereas GpsB[L35S], which is locked in the dodecameric form, fails to do so.

## GpsB interacts with and promotes lateral interactions between FtsZ polymers

We next investigated if GpsB directly interacts with polymerized FtsZ using a high-speed sedimentation assay performed with a non-hydrolyzable GTP analog (GMPCPP), which promotes the assembly of stable FtsZ polymers. In the absence of nucleotide, FtsZ was largely detected in the supernatant

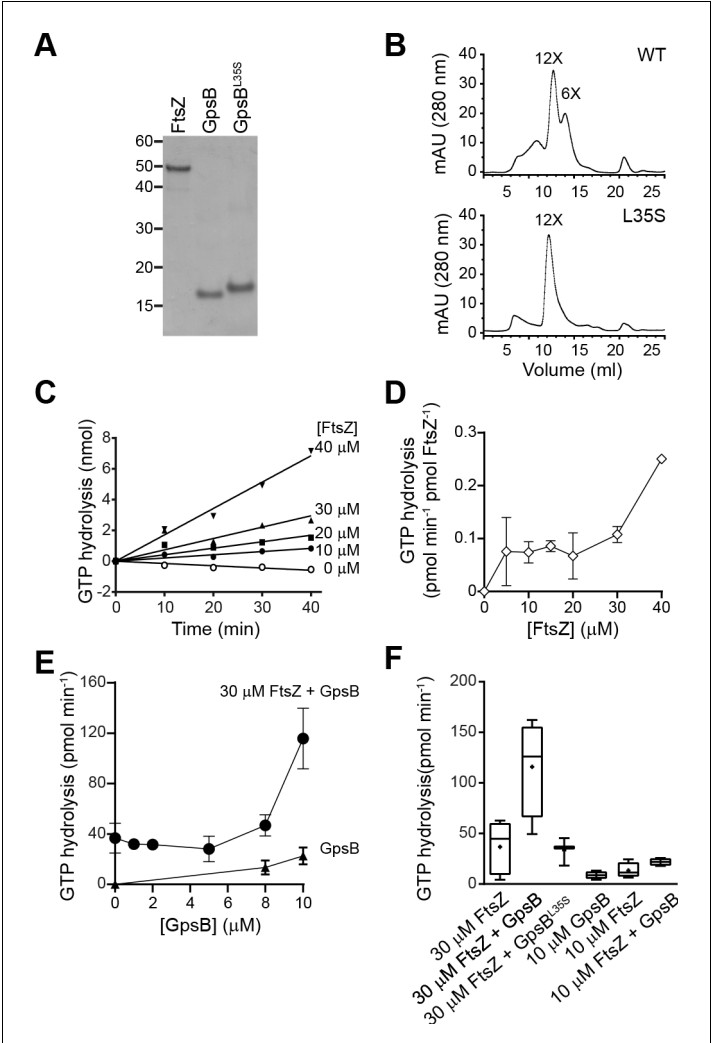

**Figure 5.** GpsB stimulates GTPase activity of FtsZ in vitro. (**A**) Coomassie-stained gel of purified FtsZ, GpsB, and GpsB$^{L35S}$. (**B**) Size exclusion chromatograms of purified GpsB (top) or GpsB$^{L35S}$ (bottom). Predicted multimerization states of the purified protein, based on migration of MW standards, is indicated above peaks (12X, dodecamer; 6X, hexamer). Shown is a representative example of at least 3 independent purifications. (**C**) Initial velocities of GTP hydrolysis by FtsZ as a function of time at various FtsZ concentrations. (**D**) GTP hydrolysis turnover rate of FtsZ as a function of FtsZ concentration. (**E**) GTP hydrolysis of increasing concentrations of GpsB alone (▲) or 30 µM FtsZ in the presence of increasing GpsB concentrations (•). Error bars represent SEM (n = 3). (**F**) Median GTP hydrolysis rates of 30 µM FtsZ and 10 µM FtsZ in the absence and presence of 10 µM GpsB or GpsB$^{L35S}$. The ends of the boxes represent the first and third quartile of measurements; bars represent the entire range of measurements; line indicates median value; '+' indicates mean value (n = 3).

DOI: https://doi.org/10.7554/eLife.38856.009

after ultracentrifugation, but in the presence of GMPCPP, more than 50% of FtsZ was detected in the pellet fraction, indicating that it had polymerized (*Figure 6A*). When GpsB was incubated with the reaction, 94% of GpsB co-sedimented with FtsZ, whereas only 20% of the nonfunctional GpsB$^{L35S}$ co-sedimented with FtsZ. Finally, in the absence of FtsZ, GpsB and GpsB$^{L35S}$ were largely soluble, suggesting that GpsB, but not GpsB$^{L35S}$, interacts with FtsZ polymers. To test if GpsB altered the ultrastructure of assembled FtsZ, we repeated the centrifugation at a slower speed to distinguish between individual or short FtsZ polymers and larger supramolecular assemblies of FtsZ. At a slower centrifugation speed, we detected 43% of FtsZ in the pellet fraction in the presence of GMPCPP (*Figure 6—figure supplement 1A*). Addition of GpsB increased the fraction of FtsZ in the

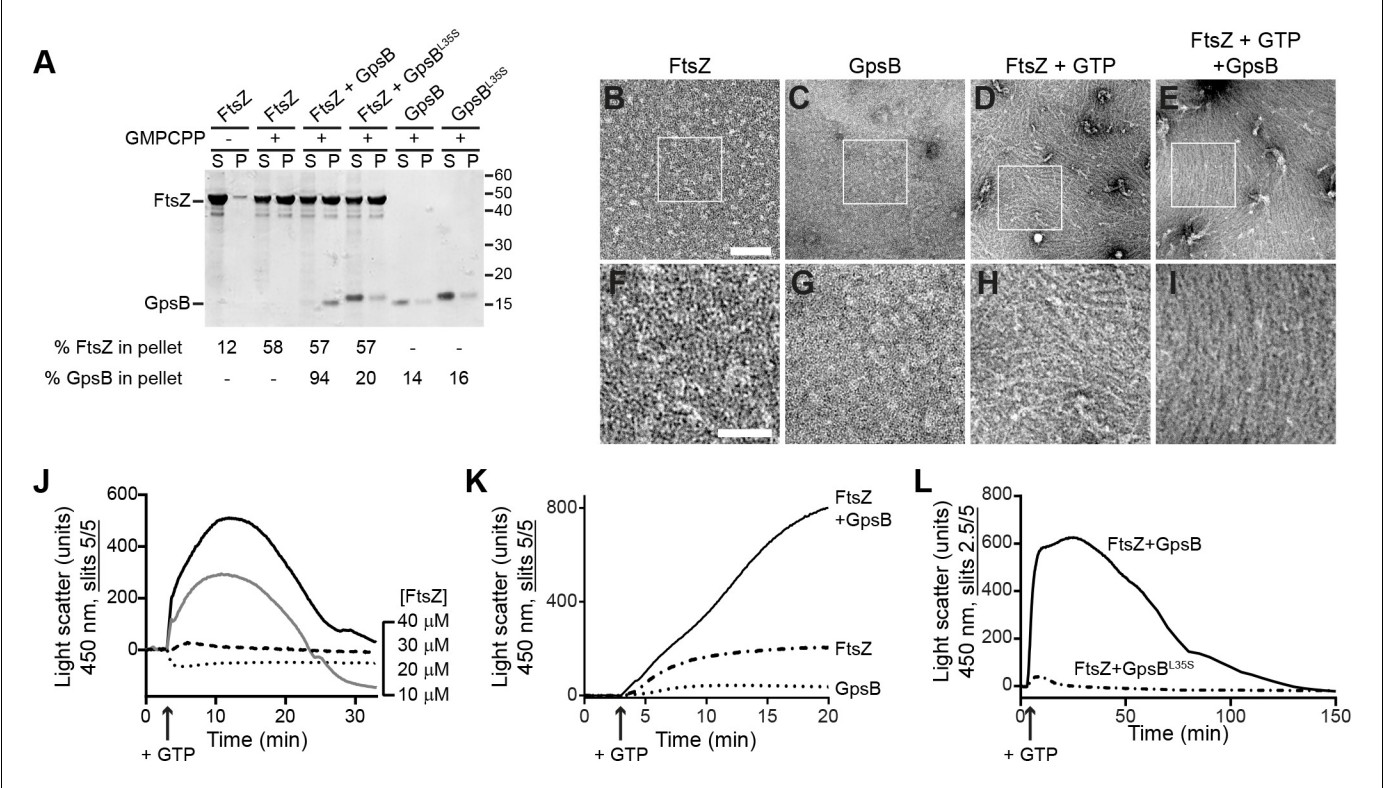

**Figure 6.** GpsB bundles FtsZ polymers in vitro. (**A**) Co-sedimentation of GpsB with polymerized FtsZ in vitro. 30 μM FtsZ were incubated in the presence or absence of GMPCPP, and 10 μM GpsB or GpsB[L35S] as indicated. Polymerized FtsZ was collected by ultracentrifugation and proteins in the resulting supernatant (**S**) and pellet (**P**) were separated by SDS-PAGE and detected by Coomassie staining. Percentage of total FtsZ or GpsB in the pellet are indicated below. Migration of MW markers are indicated to the right. Shown is a representative gel of 3 independent replicates. (**B–E**) Morphology of (**B**) purified FtsZ, (**C**) purified GpsB, (**D**) polymerized FtsZ incubated with GTP, or (**E**) polymerized FtsZ incubated with GTP and GpsB visualized using negative stain transmission electron microscopy. Scale bar: 200 nm. (**F–I**) magnified views of the areas indicated in (**B–E**), respectively. Scale bar: 100 nm. (**J**) Assembly of 10 μM (dotted trace), 20 μM (dashed), 30 μM (gray) or 40 μM (black) FtsZ measured using 90° angle light scattering. (**K**) FtsZ assembly in the presence (solid) or absence (dash-dot) of GpsB, or GpsB alone (dotted), measured by 90° angle light scattering. (**L**) Assembly and disassembly of FtsZ in the presence of limiting amount of GTP monitored by 90° angle light scattering. Time of GTP addition is indicated with an arrow. Note the difference in slit width in (**L**). Shown are representative traces of at least 3 independent experiments.

DOI: https://doi.org/10.7554/eLife.38856.010

The following figure supplements are available for figure 6:

**Figure supplement 1.** GTP hydrolysis is not required for GpsB-mediated FtsZ bundling.
DOI: https://doi.org/10.7554/eLife.38856.011

**Figure supplement 2.** Assembly of 30 μM FtsZ measured using 90° angle light scattering in the presence of GTP (red trace), GDP (blue), ATP (light green), or ADP (dark green).
DOI: https://doi.org/10.7554/eLife.38856.012

pellet to 55%, whereas addition of GpsB[L35S] did not significantly alter the pelleted fraction of FtsZ. In the presence of GTP, 29% of FtsZ was detected in the pellet fraction (*Figure 6—figure supplement 1B*), and this fraction increased to 39% in the presence of GpsB, but not GpsB[L35S]. The differential centrifugation patterns suggested that direct interaction with GpsB could alter the assembly of FtsZ polymers.

To visualize the morphology of purified FtsZ polymers with and without GpsB, we examined purified proteins in the presence and absence of GTP using negative stain transmission electron microscopy (TEM). Purified FtsZ or GpsB alone did not show any distinguishable structures by TEM (*Figure 6B–C,F–G*). In the presence of GTP, FtsZ formed linear polymers,~100 nm in length, that were abundant and scattered in different directions on the grid, indicating that it had polymerized successfully in a GTP-dependent manner (*Figure 6D,H*). In the presence GpsB and GTP, however,

FtsZ polymers formed long filaments, closer to 1 μm in length, which were oriented in a similar direction (*Figure 6E,I*). This pattern of orientation on an EM grid was reminiscent of the bundling behavior reported previously for proteins in *E. coli* that could promote lateral interactions between FtsZ filaments in vitro (*Hale et al., 2000*; *Small et al., 2007*). With GMPCPP, FtsZ polymers were very long and in the presence of GpsB also exhibited extensive lateral interactions between FtsZ filaments, indicating that GpsB-mediated bundling of FtsZ did not require GTP hydrolysis (*Figure 6— figure supplement 1C–E*).

We next monitored the kinetics of FtsZ assembly in vitro using 90° angle light scattering (*Mukherjee and Lutkenhaus, 1999*) as a function of FtsZ concentration. Incubation of either 10 μM or 20 μM purified *S. aureus* FtsZ with GTP did not result in an appreciable increase in light scattering (*Figure 6J*), consistent with the apparent high critical concentration for FtsZ assembly suggested in GTP turnover experiments (*Figure 5C–D*). Incubation of 30 μM or 40 μM FtsZ with GTP resulted in a rapid increase in light scattering that likely corresponds to the assembly of FtsZ polymers. The increase was followed by a brief plateau, likely reflecting that the reaction was at steady state, then a decrease in light scattering, corresponding to disassembly of FtsZ polymers coincident with depletion of GTP in the reaction. Such kinetics were not detected when 30 μM FtsZ was incubated with GDP, ATP, or ADP (*Figure 6—figure supplement 2A*), suggesting that the light scattering assay specifically reflects GTP-dependent dynamics of FtsZ assembly. To confirm that the decrease in light scattering corresponded to the depletion of GTP and accumulation of GDP in the reaction, we repeated the assay in the presence of a regeneration system to replenish GTP. As expected, addition of a GTP regeneration system prevented the rapid loss of scatter following the plateau (*Figure 6—figure supplement 2B*), suggesting that the decrease in *Figure 6J* represents the disassembly of FtsZ polymers as GTP becomes limiting.

Next, we tested the effect of GpsB on FtsZ assembly. Addition of 10 μM GpsB to the reaction with GTP and 30 μM FtsZ resulted in an initial increase in light scattering that was much more rapid and larger in amplitude than that of FtsZ alone in the presence of GTP (*Figure 6K*), whereas incubation of GpsB alone with GTP did not result in an increase in light scattering. To determine if the increase in light scattering due to GpsB was reversible, we followed the assembly reaction for a longer period (*Figure 6L*). Upon addition of GTP, the reaction containing FtsZ and GpsB displayed a rapid increase in light scattering, which was not observed when FtsZ was incubated with GpsB[L35S]. Note that, due to saturation of the detector in the presence of GpsB, the slit width in *Figure 6L* was adjusted, precluding a direct comparison between the signal amplitudes shown in *Figure 6J K*. After reaching a plateau, the reaction containing WT GpsB displayed a steady decrease in light scattering, suggesting that the assembly of the higher order FtsZ structures generated in the presence of GpsB was reversible, in contrast to the behavior of other FtsZ bundling proteins reported in other systems. We therefore conclude that GpsB directly interacts with polymerized FtsZ and bundles FtsZ filaments. Taken together with the observation that GpsB also triggers GTP hydrolysis by FtsZ, we propose that FtsZ bundling by GpsB increases FtsZ local concentration and triggers GTP hydrolysis which, in the light scattering assay, is linked to the disassembly of FtsZ polymers as GTP is depleted.

## Discussion

Since binary fission has been traditionally studied in rod-shaped model organisms, the roles of factors that participate in cell division of spherical bacteria have been less well characterized (*Eswara and Ramamurthi, 2017*). In this report, we investigated the role of a coiled-coil protein, GpsB, during cell division in the spherical bacterium *S. aureus*. Unlike the orthologs of GpsB in other systems, we report that GpsB directly interacts with FtsZ, the core component of the bacterial cell division machinery and increases the GTPase activity of FtsZ. We also demonstrate that GpsB promotes bundling of FtsZ filaments in vitro. We propose a model in which the bundling of *S. aureus* FtsZ by GpsB raises the local concentration of FtsZ transiently so that it may robustly hydrolyze GTP, and thereby participates in remodeling the constricting divisome during cytokinesis. A recent report suggested that cell division in *S. aureus* proceeds in two principal steps: an initial FtsZ treadmilling-dependent step in which membrane invagination initiates, followed by recruitment of peptidoglycan remodeling enzymes by later arriving divisome components that mediates the progression and completion of cell division (*Monteiro et al., 2018*). We propose that GpsB may participate in the initial step that stabilizes FtsZ at mid-cell and activates GTP hydrolysis (by increasing the local

concentration of FtsZ via a bundling-like mechanism) to trigger FtsZ treadmilling, which is linked to constriction of the membrane and concurrent peptidoglycan synthesis (*Bisson-Filho et al., 2017*; *Yang et al., 2017*).

In our model, FtsZ requires a high concentration to polymerize and to hydrolyze GTP efficiently. In the presence of GpsB, though, FtsZ filaments are laterally bridged to form higher order supramolecular structures (*Figure 7A*). Intracellular levels of FtsZ and GpsB are reported to be approximately 4452 and 1659 molecules per cell, respectively (*Zühlke et al., 2016*) (S. Fuchs, personal communication). Considering a cell diameter of 0.8 µm, this equates to intracellular concentrations of 28 uM FtsZ and 10 uM GpsB and corresponds closely with our in vitro reaction conditions.

We show that GpsB is a multimer and propose that it may harbor 6–12 binding sites per multimer to recruit and bridge multiple FtsZ proteins. We propose that the bridging of FtsZ filaments also serves to increase the local FtsZ concentration and enhances GTP hydrolysis. Unlike other proteins that bundle FtsZ irreversibly in vitro by inhibiting GTP hydrolysis, incubation with GpsB ultimately allows for the subsequent disassembly of FtsZ polymers once GTP is depleted (*Figure 7A*). To our knowledge, this is the first report of an FtsZ regulatory protein that promotes both lateral interactions between FtsZ filaments while also stimulating GTP hydrolysis. Furthermore, considering the intracellular concentrations of FtsZ and GpsB and what we have observed biochemically, FtsZ polymers and regulators appear poised at the threshold between assembly and disassembly, enabling tight control over the division process.

Our view is supported by multiple lines of evidence. First, overexpression of *gpsB* resulted in the enlargement of *S. aureus* cells, reminiscent of the phenotype caused by depletion of FtsZ (*Pinho and Errington, 2003*), likely due to increased FtsZ GTPase activity leading to the inability of FtsZ to polymerize and treadmill in a concerted fashion. Curiously, overexpression of *S. aureus gpsB* in *B. subtilis*, but not the *B. subtilis gpsB* ortholog, resulted in filamentation, suggesting that *S. aureus* GpsB harbors a unique cell division-modulating activity that is not exhibited by the *B. subtilis*

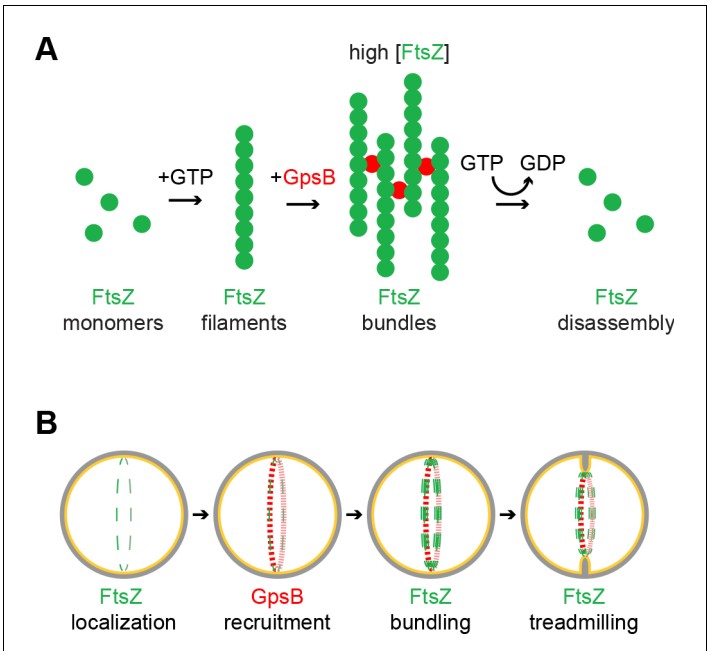

**Figure 7.** Model of GpsB remodeling of FtsZ in *S.aureus*. (**A**) Molecular model. FtsZ (green) filaments, which form upon GTP binding, directly interact with GpsB (red). Filament-bound GpsB molecules promote lateral interactions between FtsZ filaments, thereby raising the local concentration of FtsZ, which drives GTP hydrolysis that leads to FtsZ disassembly. (**B**) Cellular model. FtsZ (green) ring localizes at mid-cell and recruits GpsB (red), which initially drives lateral interactions between FtsZ filaments to promote Z-ring stabilization at that position. Subsequent stimulation of FtsZ GTP hydrolysis, caused by a local increase in FtsZ concentration, stimulates FtsZ treadmilling which is linked to membrane constriction and concurrent peptidoglycan synthesis at mid-cell.
DOI: https://doi.org/10.7554/eLife.38856.013

version. Second, depletion of GpsB in *S. aureus* resulted in the arrest of cytokinesis and abrogation of initiation of cell division. Third, we observed that GpsB co-localized with the cell division machinery at the onset of cytokinesis, and co-constricted with the invaginating membrane during cell division, consistent with its role in modulating the activity of FtsZ. Fourth, we found that purified GpsB directly interacted with FtsZ in vitro and stimulated the GTPase activity of FtsZ, consistent with the ability of GpsB to inhibit cell division in vivo when overproduced. Finally, when incubated with FtsZ in vitro, GpsB promoted lateral interactions between FtsZ polymers, but also allowed for the ultimate disassembly of FtsZ in vitro once GTP had been depleted.

Our genetic, cytological, and biochemical data in sum suggest a model in which *Staphylococcal* FtsZ begins assembling at mid-cell and recruits GpsB to that location (*Figure 7B*) where GpsB stabilizes the Z-ring via a bundling-like mechanism that concentrates and organizes FtsZ polymers. We propose that this reinforces the faithful and robust assembly of FtsZ at mid-cell at the onset of cell division and drives an increase in the local concentration of FtsZ, which stimulates its GTPase activity, which is linked to treadmilling- an activity that is likely coincident with the initial membrane constriction that initiates cytokinesis. After completion of cytokinesis, GpsB redistributes to the cell periphery and awaits the next round of cell division. It is tempting to speculate that this dynamic redistribution of GpsB, presumably coincident with a dynamic ability to modulate FtsZ activity, is dependent on the multimerization state of GpsB. In this way, the hexameric and dodecameric populations of purified GpsB could represent the active and inactive forms, respectively, of the protein that may mediate its interaction with FtsZ. Consistent with this model, the inactive GpsB[L35S] was locked in the dodecameric form and did not exhibit the dynamic redistribution from the cell periphery to the cell division site in vivo. Identifying the factors that regulate the multimerization state of GpsB could therefore provide an understanding into the temporal regulation of cell division in *S. aureus*. Interestingly, depletion of a known interaction partner of GpsB, EzrA, also leads to cell enlargement, hinting at a possible collaboration between these two proteins in regulating cell division (*Jorge et al., 2011*; *Steele et al., 2011*).

Several divisome proteins in *E. coli* and *B. subtilis* that positively regulate cell division by bundling FtsZ polymers do so via inhibition of FtsZ GTPase activity (*Durand-Heredia et al., 2011*; *Hale et al., 2011*; *Lutkenhaus et al., 2012*; *Mohammadi et al., 2009*; *Pacheco-Gómez et al., 2013*; *Singh et al., 2008*; *Small et al., 2007*; *Tsang and Bernhardt, 2015*). In our model, the seemingly contradictory observation that GpsB stimulates GTP hydrolysis, even though it promotes FtsZ bundling may be explained by the proposition that FtsZ bundling is not the ultimate activity of GpsB. Rather, we envision that FtsZ bundling is an intermediate step that increases local FtsZ concentration to stimulate GTP hydrolysis (*Figure 7A*). This set of opposing activities exhibited in *S. aureus* in a single protein is reminiscent of the model in *E. coli*, where FtsZ polymers bundled by other proteins require a separate protein, FtsA, that can disrupt the bundles and destabilize FtsZ polymers (*Conti et al., 2018*; *Krupka et al., 2017*). In a slight variation of this model, since FtsZ bundling in *B. subtilis* requires C-terminal positively charged residues (*Buske and Levin, 2012*), it is conceivable that GpsB modulates exposure of the C-terminus of FtsZ to promote FtsZ self-interactions or remodels FtsZ to stabilize a conformation associated with enhanced GTP hydrolysis. In this way, GpsB, an essential *S. aureus* protein, may orchestrate the organization, stabilization, and activity of FtsZ to remodel the divisome during cell division.

# Materials and methods

**Key resources table**

| Reagent type (species) or resource | Designation | Source or reference | Identifiers | Additional information |
|---|---|---|---|---|
| Strain, strain background (*Bacillus subtilis*) | PY79 | *Youngman et al. (1984)* | | Wild type |
| Strain, strain background (*Bacillus subtilis*) | GG7 | This paper | | *amyE::Phyperspank -gpsB[Sa] spec* |

*Continued on next page*

Continued

| Reagent type (species) or resource | Designation | Source or reference | Identifiers | Additional information |
|---|---|---|---|---|
| Strain, strain background (*Bacillus subtilis*) | GG8 | This paper | | *amyE::Phyperspank-gpsB^{Sa}-gfp spec* |
| Strain, strain background (*Bacillus subtilis*) | GG35 | This paper, derived from FG345, **Gueiros-Filho and Losick (2002)** | | Δ*ezrA::spec::erm amyE::Phyperspank-gpsBSa spec* |
| Strain, strain background (*Bacillus subtilis*) | CS26 | This paper, derived from BKE22320 (BGSC) | | Δ*ponA::erm amyE::Phyperspank-gpsB^{Sa} spec* |
| Strain, strain background (*Bacillus subtilis*) | CS24 | This paper, derived from BKE15770 (BGSC) | | Δ*prkC::erm amyE::Phyperspank-gpsB^{Sa} spec* |
| Strain, strain background (*Bacillus subtilis*) | CS40 | This paper | | Δ*gpsB::tet amyE::Phyperspank-gpsB^{Sa} spec* |
| Strain, strain background (*Bacillus subtilis*) | CS94 | This paper, derived from KR546, **Ramamurthi and Losick (2009)** | | Δ*divIVA::erm amyE::Phyperspank-gpsB^{Sa} spec* |
| Strain, strain background (*Bacillus subtilis*) | GG9 | This paper, derived from AD3007, **Eswaramoorthy et al. (2011)** | | *amyE::Phyperspank-gpsB^{Sa} spec ftsAZΩftsAZ-gfp erm* |
| Strain, strain background (*Bacillus subtilis*) | GG18 | This paper | | *amyE::Phyperspank-gpsB^{Bs} spec* |
| Strain, strain background (*Bacillus subtilis*) | GG19 | This paper | | *amyE::Phyperspank-gpsB^{Bs}-gfp spec* |
| Strain, strain background (*Bacillus subtilis*) | PE448 | This paper | | *amyE::Phyperspank-gpsB^{Sa}-L35S-gfp spec* |
| Strain, strain background (*Staphylococcus aureus*) | SH1000 (aka PL3055) | **Horsburgh et al. (2002)** | | Wild type |
| Strain, strain background (*Staphylococcus aureus*) | SH1000 pCL15 | **Luong and Lee (2006)** | | *bla cat* |
| Strain, strain background (*Staphylococcus aureus*) | SH1000 pPE45 | This paper | | pCL15 backbone, Pspac-gpsBSa bla cat |
| Strain, strain background (*Staphylococcus aureus*) | SH1000 pPE83 | This paper | | pCL15 backbone, Pspac-gpsBBs bla cat |
| Strain, strain background (*Staphylococcus aureus*) | SH1000 pPE79 | This paper | | pCL15 backbone, $P_{spac}$-gpsB^{Sa-L35S} bla cat |
| Strain, strain background (*Staphylococcus aureus*) | SH1000 pPE46 | This paper | | pCL15 backbone, $P_{spac}$-gpsB^{Sa}-gfp bla cat |
| Strain, strain background (*Staphylococcus aureus*) | SH1000 pPE80 | This paper | | pCL15 backbone, $P_{spac}$-gpsB^{Sa-L35S}-gfp bla cat |
| Strain, strain background (*Staphylococcus aureus*) | SH1000 pEPSA5 | **Forsyth et al. (2002)** | | *bla cat* |
| Strain, strain background (*Staphylococcus aureus*) | SH1000 pGG59 | This paper | | pEPSA5 backbone, $P_{xyl}$-gpsB^{antisense} bla cat |

*Continued on next page*

*Continued*

| Reagent type (species) or resource | Designation | Source or reference | Identifiers | Additional information |
|---|---|---|---|---|
| Strain, strain background (*Staphylococcus aureus*) | SH1000 pRB42 | This paper | | pJB67 backbone, $P_{Cd}$-zapA$^{Sa}$-gfp bla erm |
| Sequence-based reagent (oligonucleotide) | oP36 | This paper | | AAAAAGCTTACATAA GGAGGAACTA CTATGTCAGATGTTTCATT GAAATTATCAGCA |
| Sequence-based reagent (oligonucleotide) | oP37 | This paper | | AAAGCTAGCTTTACCA AATACAG CTTTTTCTAAGTTTGA |
| Sequence-based reagent (oligonucleotide) | oP38 | This paper | | AAAGCATGCTTATTTACCAAAT ACAGCTTTTTCTAAGTTTGA |
| Sequence-based reagent (oligonucleotide) | oP46 | This paper | | AAAGCTAGCATGAGTAAAG GAGAAGAACTTTTC |
| Sequence-based reagent (oligonucleotide) | oP24 | This paper | | GCCGCATGCTTATTTGTA TAGTTCATCCATGCC |
| Sequence-based reagent (oligonucleotide) | oP100 | This paper | | AAAGTCGACACATA AGGAGGAACTACTATGCTTGCTGAT AAAGTAAAG CTTTCTGCG |
| Sequence-based reagent (oligonucleotide) | oP101 | This paper | | AAAGCTAGCATCA TAAAGCTTG CTGCCAAAAACGTG |
| Sequence-based reagent (oligonucleotide) | oP102 | This paper | | AAAGCTAGCTCAAT CATAAAGC TTGCTGCCAAAAACGTG |
| Sequence-based reagent (oligonucleotide) | oP195 | This paper | | AAAGGATCCTCAATCATAAAG CTTGCTGCCAAAAACGTG |
| Sequence-based reagent (oligonucleotide) | oP187 | This paper | | AAAGAATTCTTATTTACCA AATACAGCTTTTTC TAAGTTTGAAATACGTTTTA AAATATCTAC |
| Sequence-based reagent (oligonucleotide) | oP188 | This paper | | AAAGGATCCGAGG TGGAAAAAATGT CAGATGTTTCATTGAA ATTATCAGC |
| Sequence-based reagent (oligonucleotide) | oP236 | This paper | | AAAGTCGACTAATGAGGAG GAAAAAATGACACAGTTTAAAAAC AAGGTAAATGTATCAATTAATGATCAG |
| Sequence-based reagent (oligonucleotide) | oP237 | This paper | | AAAGCTAGCCGCTGCTG CAATTTGT GAATTTGTTGTTTCAAACGT |
| Sequence-based reagent (oligonucleotide) | oP47 | This paper | | AAAGGATCCTTATTTGTATAGT TCATCCATGCC |
| Antibody | anti-GpsB | This paper | | Raised against purified GpsB-His$_6$ |
| Antibody | anti-SigA | Ramamurthi lab | | Raised against purified *B. subtilis* SigA-His$_6$ |

## Strain construction and general methods

*B. subtilis* strains used in this study are derivatives of PY79 (*Youngman et al., 1984*), and *S. aureus* strains are derivatives of SH1000 (*Horsburgh et al., 2002*). To overproduce GpsB or GpsB-GFP orthologs in *B. subtilis*, *gpsB* (HindIII/SphI; primers oP36/oP38, please see Key Resources Table for

primers) or *gpsB-gfp* (HindIII/NheI; oP36/37 for *gpsB* without stop codon; and NheI/SphI; oP46/24 for *gfp* with stop codon) were PCR amplified and cloned into the 5′ *Hind*III and 3′ *Nhe*I restriction sites in pDR111 (D. Rudner) to place it under control of the isopropyl β-D-1-thiogalactopyranoside (IPTG)-inducible $P_{hyperspank}$ promoter. The resulting plasmids (pGG3, *gpsB*; pGG4, *gpsB-gfp*) were integrated into the *amyE* locus in the *B. subtilis* chromosome by double recombination. Similarly, *B. subtilis gpsB* was constructed using primers oP100/102 (*gpsB*; SalI/NheI) and *gpsB-gfp* was constructed by ligating the products of oP100/101 (*gpsB* no stop codon; SalI/NheI) and oP46/24 (*gfp*; NheI/SphI). To produce *S. aureus* GpsB or GpsB-GFP in *S. aureus*, *gpsB* or *gpsB-gfp* were PCR amplified and cloned into the 5′ HindIII and 3′ SphI restriction sites in the pCL15 plasmid (*Luong and Lee, 2006*), downstream of the IPTG-inducible $P_{spac}$ promoter, to create pPE45 and pPE46, respectively. The L35S substitution was introduced using the QuikChange Site-Directed Mutagenesis kit (Agilent) using pPE45, pPE46, pGG3, or pGG4 as a template. To express *B. subtilis gpsB* in *S. aureus*, a pCL15-based vector pPE83 was constructed by amplifying and inserting the *B. subtilis gpsB* fragment with the help of primer pairs oP100/195 (SalI/BamHI). To express the anti-sense RNA of the *gpsB* open reading frame and ribosome binding site under control of a xylose-inducible promoter, using primers oP187/188 abutted by *EcoR*I and *BamH*I restriction sites and cloned into plasmid pEPSA5 (*Forsyth et al., 2002*) to create plasmid pGG59. Plasmid pRB42 (*zapA-gfp*) was constructed using primers oP236/237 (*zapA* no stop codon; SalI/NheI) and oP46/47 (*gfp*; NheI/BamHI) and inserted into cadmium-inducible plasmid pJB67 (*Windham et al., 2016*). Plasmids were first introduced into *S. aureus* RN4220 by electroporation, then transduced into strain SH1000. Expression was induced by addition of 1 mM IPTG or 1% xylose or 1.25 μM $CdCl_2$, as required, in the growth medium.

## Cell lysates

For immunoblot analysis of cell extracts, overnight cultures of *S. aureus* were diluted 1:50 into 10 ml tryptic soy broth (TSB) and were grown to mid-logarithmic phase, harvested by centrifugation, and resuspended in 1 ml buffer A (see below) containing 200 mM KCl, 1 mM dithiothreitol, and 10 mg/ml lysostaphin and incubated for 15 min at room temperature. Suspensions were then sonicated (3 intervals at 10 s each at 20% power level), then cleared by centrifugation at 14,000 × g for 10 min. Supernatants were isolated and centrifuged at 100,000 × g for 1 hr to separate soluble (supernatant) fraction from insoluble (pellet) fraction. Supernatants were removed for analysis. Pellets were resuspended in 1 ml buffer (no lysostaphin) containing 0.01% SDS. Samples were separated using 8–16% SDS-PAGE (BioRad), transferred to nitrocellulose membrane, and probed with rabbit antisera raised against purified *S. aureus* GpsB or *B. subtilis* SigA antibody.

## Microscopy

Overnight *B. subtilis* cultures grown at 22°C in Luria-Bertani (LB) medium were diluted 1:20 into fresh LB medium and grown for 2.5 hr at 37°C. Overnight cultures of *S. aureus* in TSB, containing 15 μg/ml chloramphenicol and/or 5 μg/ml erythromycin for plasmid maintenance if necessary, were diluted into fresh medium and grown to mid-logarithmic phase. 1 mM IPTG was added as required for 3 hr. 1 ml cultures were washed with PBS and resuspended in ~100 μl PBS containing 1 μg/ml fluorescent dye FM4-64 and/or 2 μg/ml DAPI to visualize membranes and DNA, respectively. 5 μl was spotted on a glass bottom culture dish (Mattek) and covered with a 1% agarose pad made with distilled water and imaged at 25°C. For time lapse, a 5 μl aliquot of SH1000 pGG59 cells grown in TSB/chloramphenicol until mid-log phase was spotted on a glass bottom culture dish and covered with an agarose pad made with TSB/chloramphenicol containing 1% xylose to induce expression of the *gpsB* antisense RNA. After 20 min of equilibration in the microscopy environmental chamber, images were obtained at 15 min intervals for 4 hr at 25°C. For FtsZ inhibition experiments, mid-log phase cells were incubated with 2 μg/ml PC190723 and samples for imaging were collected after 3 hr. Cells were viewed with a DeltaVision Core microscope system (Applied Precision/GE Healthcare) equipped with a Photometrics CoolSnap HQ2 camera and an environmental chamber. Seventeen planes for standard microscopy and four planes for time-lapse microscopy were acquired every 200 nm, and the data were deconvolved using SoftWorx software as described previously (*Tan et al., 2015*). For structured illumination microscopy, cells were viewed using a DeltaVision OMX (Applied

Precision/GE Healthcare) comprising an OMX optical microscope (version 4), equipped with a sCMOS camera.

## FtsZ and GpsB purification

To purify FtsZ, *S. aureus ftsZ* was PCR amplified and cloned into the pET28a(+) vector (EMD Millipore) using 5' *Nde*I and 3' *Xho*I restriction sites, resulting in the addition of an N-terminal histidine tag followed by a thrombin cleavage site. Expression was induced in BL21(λDE3)::Δ*clpP* cells grown in LB broth supplemented with 50 µg/ml Kanamycin for plasmid maintenance, at 30°C by adding 1 mM IPTG after cells reached an optical density (600 nm) of 1.0. Cells were harvested by centrifugation, resuspended in buffer A [20 mM HEPES (pH 7.5), 50 mM KCl, 5 mM $MgCl_2$ and 10% glycerol], and lysed by French press. Soluble extract was collected by centrifugation at 30,000 × g for 30 min at 4°C and applied to an IMAC column (TALON Superflow, GE Healthcare), and washed with Buffer A containing 10 mM imidazole. Untagged FtsZ was eluted with thrombin (4 U; Novagen) and then 0.5 mM phenylmethylsulphonyl fluoride was added to inactivate thrombin.

To purify GpsB-$His_6$, *S. aureus gpsB* was PCR amplified and cloned into pET28a(+) using 5' *Xba*I and 3' *Bam*HI restriction sites, using primers to append a $His_6$ tag to the C-terminus. Overproduction of $GpsB^{Sa}$-$His_6$ in *B. subtilis* resulted in cell filamentation similar to untagged $GpsB^{Sa}$, suggesting that the His6-tagged protein was functional. The L35S substitution was introduced using the Quik-Change Site-Directed Mutagenesis kit (Agilent). Expression was induced in BL21(λDE3)::Δ*clpP* cells grown in LB broth supplemented with 25 µg/ml Kanamycin for plasmid maintenance, at 37°C by adding 0.5 mM IPTG for 2 hr after cells reached an optical density (600 nm) of 0.6. Cells were harvested by centrifugation and resuspended in 30 ml cold buffer B [50 mM sodium phosphate (pH 8.0), 500 mM NaCl, 20 mM imidazole, 1 mM EDTA, 10% Glycerol, 3 mM DTT] and lysed by sonication (5 s on/10 s off cycle for 5 min). Lysate was cleared by centrifugation for 30 min at 40,000 × g; cleared lysate was passed through a $Ni^{2+}$-NTA column equilibrated with buffer B, washed with 20 column volumes buffer B, and eluted with buffer B containing 200 mM imidazole. Imidazole was removed with a PD10 desalting column and eluted with buffer A containing 250 mM KCl and 1 mM DTT. To ensure the final buffer composition the protein was dialyzed over night at 4°C against buffer A.

## FtsZ Assembly and GTP hydrolysis

FtsZ assembly was monitored by 90° angle light scattering using an Agilent Eclipse fluorescence spectrophotometer with excitation and emission wavelengths set to 450 nm and slit widths of 5/5 or 2.5/5, where indicated. FtsZ (30 µM) was added to reactions (80 µl) containing assembly buffer (20 mM HEPES pH 7.5, 140 mM KCl, 5 mM $MgCl_2$) with and without GpsB or $GpsB^{L35S}$ (1 or 10 µM), where indicated. Baseline readings were collected for 3 min, 2 mM GTP was added and light scattering was measured for up to 300 min. GMPCPP-stabilized FtsZ polymers were assembled by incubating FtsZ (30 µM) with 0.5 mM GMPCPP in the absence and presence of GpsB or $GpsB^{L35S}$ (10 µM) for 10 min and collected by centrifugation either for 30 min at 129,000 x *g* (*Figure 6A*), 20 min at 20,000 x *g*, or 20 min at 90,000 x *g* (Figure 2—figure supplement 1A–B), as indicated. Where indicated, polymerization was stimulated with GTP (2 mM) and a nucleotide regenerating system containing acetate kinase (25 µg ml$^{-1}$) and acetyl phosphate (15 mM) was included to prevent GDP accumulation. Supernatants and pellets were resuspended in equivalent volumes of LDS sample buffer (Life Technologies) and analyzed by SDS-PAGE and Coomassie staining. The relative amounts of FtsZ, GpsB and $GpsB^{L35S}$ in supernatant and pellet fractions were quantified by densitometry using ImageJ (NIH).

FtsZ GTP hydrolysis activity was monitored by detection of free phosphate using Biomol Green (Enzo Life Sciences). Reactions containing FtsZ (0–40 µM) in the absence and presence of GpsB and GpsB(L35S) (0–10 µM) were incubated with 2 mM GTP in assembly buffer at room temperature. Phosphate was measured at 0 and 15 min by comparison to a phosphate standard curve. Rates were calculated by measuring the amount of free phosphate released during the incubation period. At low FtsZ concentrations, reactions were incubated for 60 min.

## Electron microscopy

FtsZ (30 µM) polymers were assembled in buffer (20 mM HEPES pH 7.5, 140 mM KCl, 5 mM $MgCl_2$) in the presence or absence of GpsB (10 µM) by addition of 2 mM GTP. After 10 min, reactions were applied to formvar/carbon coated 300 mesh grids, fixed with 2.5% glutaraldehyde in 0.15M sodium cacodylate buffer (pH 7.4) and stained with 2% aqueous uranyl acetate. Samples were imaged by transmission electron microscopy using a FEI Tecnai G2 Spirit BioTWIN 80Kv instrument equipped with a SIS Morada 11 Megapixel camera.

## Acknowledgements

We thank S Gottesman, S Wickner, M Maurizi, and D Chattoraj for suggestions; V Lee and members of our labs for comments on the manuscript; JP Cooper's lab (NCI) for use of their SIM microscope; H Arjes and P Levin for *S. aureus* strain SH1000 and plasmid pEPSA5; L Shaw for *S aureus* plasmid pJB67; D Ziegler (*Bacillus* Genetic Stock Center) for various *B. subtilis* strains; and Marc Llaguno and Xinran Liu at the Center for Cellular and Molecular Imaging at Yale School of Medicine for TEM. This work was funded by a start-up grant from the University of South Florida (PJE); the National Institute of General Medical Sciences of the National Institutes of Health #R01GM118927 (JLC), #R01GM128037 (PJE), and #SC2 GM105419 (KMT); USDA National Institute of Food and Agriculture, Hatch project #232838 (JLC); Howard University Medical Alumni Association (KMT), supported in part by: Howard University Research Centers in Minority Institutions, funded by the National Institute on Minority Health and Health Disparities (G12MD007597), NIH; and the Intramural Research Program of the NIH, National Cancer Institute, Center for Cancer Research (KSR).

## Additional information

### Funding

| Funder | Grant reference number | Author |
| --- | --- | --- |
| National Institutes of Health | R01GM128037 | Prahathees J Eswara |
| University of South Florida | Start-up grant | Prahathees J Eswara |
| National Institutes of Health | G12MD007597 | Karl Thompson |
| National Institutes of Health | SC2 GM105419 | Karl Thompson |
| U.S. Department of Agriculture | 232838 | Jodi Camberg |
| National Institutes of Health | R01GM118927 | Jodi Camberg |
| National Institutes of Health | Intramural Research Program | Kumaran S Ramamurthi |

The funders had no role in study design, data collection and interpretation, or the decision to submit the work for publication.

### Author contributions

Prahathees J Eswara, Conceptualization, Formal analysis, Supervision, Funding acquisition, Investigation, Methodology, Writing—original draft, Project administration; Robert S Brzozowski, Marissa G Viola, Gianni Graham, Catherine Spanoudis, Catherine Trebino, Jyoti Jha, Joseph I Aubee, Formal analysis, Investigation, Writing—review and editing; Karl M Thompson, Jodi L Camberg, Formal analysis, Funding acquisition, Investigation, Project administration, Writing—review and editing; Kumaran S Ramamurthi, Conceptualization, Formal analysis, Funding acquisition, Writing—original draft, Project administration

### Author ORCIDs

Prahathees J Eswara https://orcid.org/0000-0003-4430-261X
Kumaran S Ramamurthi http://orcid.org/0000-0002-2335-3568

## Decision letter and Author response

Decision letter https://doi.org/10.7554/eLife.38856.017
Author response https://doi.org/10.7554/eLife.38856.018

---

## Additional files

### Supplementary files

• Transparent reporting form
DOI: https://doi.org/10.7554/eLife.38856.014

### Data availability

All data generated or analysed during this study are included in the manuscript and supporting files.

---

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
