## [Decision Letter]

[Editors’ note: a previous version of this study was rejected after peer review, but the authors submitted for reconsideration. The first decision letter after peer review is shown below.]

Thank you for submitting your work entitled "An essential *Staphylococcus aureus* cell division protein regulates assembly and disassembly of FtsZ" for consideration by *eLife*. Your article has been reviewed by three peer reviewers, and the evaluation has been overseen by a Reviewing Editor and a Senior Editor. The reviewers have opted to remain anonymous.

Our decision has been reached after consultation between the reviewers. Based on these discussions and the individual reviews below, we regret to inform you that your work will not be considered further for publication in *eLife*.

While the three reviewers note that discovering the function of GpsB in cell division would be an important finding for the field, all feel that the study it is not sufficiently advanced to support the conclusions. In particular, effects other than interference with FtsZ polymerization could explain the cell biology phenotypes, and the biochemical characterization of GpsB is insufficiently advanced to justify the conclusions. Specifically, the reviewers do not feel that the in vitro data show unequivocally that GspB is an FtsZ-bundling factor and alternative explanations are possible. Last, the proposed constriction model is at odds with recent work suggesting that FtsZ might rather distribute peptidoglycan synthesis along the septal ring. Given the extent of the requested modifications, we have no choice but to reject the manuscript. The individual reviews are provided below to help you prepare the manuscript for another journal.

Reviewer #1:

In this manuscript, the authors analyze the function of the GpsB protein in *S. aureus*. GpsB and its paralog DivIVA are well-studied in *Firmicutes* and *Actinobacteria*. In these species, these two proteins have been shown to be involved in regulation of cell growth, regulation of cell division and/or in chromosome segregation. Here, it is reported that overproduction of GpsB from *S. aureus* in *B. subtilis* gives rise to filamentous cells; overproduction of GpsB in *S. aureus* gives rise to large spherical cells; depletion of GpsB in *S. aureus* results in inhibition of cell division; a gspB mutation is isolated that does not affect cell shape upon expression in *B. subtilis*; GpsB localization is determined and found to be dynamic over the cell cycle: and, that GpsB interacts with FtsZ in vitro.

Previously, it has been reported that GpsB is essential in *S. aureus*. This manuscript aims to clarify the function of GpsB in *S. aureus*. The data presented present a good starting point for this clarification; however, as outlined below I have several major problems with the manuscript that will require additional experiments. In its current format, the data do not support the major conclusions.

1) In the Introduction it would be very helpful if an explanation was included for how DivIVA and GpsB proteins are defined and how they differ from each other.

GpsB expression in *B. subtilis*:

It is not clear that GpsB from *S. aureus* and *B. subtilis* accumulate at the same levels in the strains were the two proteins are overproduced.

The multiple nucleoids in the filamentous *B. subtilis* cells formed after overproduction of GpsB from *S. aureus* are not clearly segregated. Therefore, it is not clear what actually causes the filamentous phenotype, i.e. does GpsB from *S. aureus* interfere with chromosome segregation or with cell division or both processes?

Does overproduction of GpsB from *S. aureus* cause a filamentous phenotype in a *B. subtilis* strain that lacks the native GpsB or DivIVA?

GpsB overproduction in *S. aureus*:

Please provide numbers for how many uninduced cells have septa and how many cell shave septa after induction of GpsB overproduction.

The large cells formed after GpsB overproduction seem to be devoid of nucleoid. So more quantitative analyses using chromosomal markers need to be included to clearly demonstrate that chromosome segregation is normal.

From the data provided it is not clear if overproduced GpsB interferes with chromosome segregation, cell division, cell growth or a combination of these processes. It is essential to include additional experiments to precisely define the process that overproduced GpsB interferes with is precisely defined.

GpsB depletion in *S. aureus*:

Please include experimental data to demonstrate that GpsB has been depleted in the experiments shown in Figure 2. Along the same lines, it seems that a construct in which expression of gpsB is regulated directly (and without the extra step with the antisense RNA) would be a more simple experimental setup.

Only a few *S. aureus* cells seem to be affected in their cell shape in the depletion experiment shown in Figure 2A, B. More quantitative data are needed.

From the experiments in Figure 2B it is by no means clear that chromosome segregation proceeds normally. In fact, it looks as if segregation has been halted. So, more quantitative analyses using chromosomal markers need to be included to clearly demonstrated that chromosome segregation is normal.

As is the case for the overproduction experiment, it is not clear if the depletion phenotype is caused by defects in chromosome segregation, cell division, cell growth or a combination of these processes. It is essential to include additional experiments to precisely define the process that is affected upon GpsB depletion.

GpsB^L35S^:

What is the phenotype of the gpsB^L35S^ mutant in *S. aureus*?

Cell biology:

The experiments nicely demonstrate that GpsB localize to the septum. Does this localization depend on FtsZ, i.e. what happens to GpsB localization upon depletion of FtsZ?

To establish a more direct connection between cell division and GpsB it is essential to include experiments on Z-ring formation and localization in the strains overproducing GpsB or depleted for GpsB.

in vitro experiments:

Please use the same units in Figure C, D and E.

All experiments in Figure 5: Please include EM images to unequivocally demonstrate that FtsZ forms filaments and that these filaments are altered by GpsB.

All experiments in Figure 5: Please include control experiments that the effect of GTP on FtsZ polymerization is specific to GTP.

Reviewer #2:

Eswara et al. describe analysis of the *S. aureu*s homolog of GpsB, a protein implicated in varied aspects of cell division among the firmicutes. Using several approaches they come to the conclusion that in contrast to other bacteria, Sa GpsB interacts directly with the highly conserved cell division protein FtsZ to control its activity. I was initially enthusiastic about this study and its potential to define the function of GpsB, a protein whose true nature has been the source of significant debate in the field. Unfortunately, many of the experiments presented in this paper are poorly designed and the physiological relevance of the authors' findings is unclear. Experiments investigating the impact of GpsB on *B. subtilis* growth and division, the apparent inclusion of an N-terminal His tag on Sa FtsZ in sedimentation and light scattering experiments, and the extremely long duration of the light scattering assays (3 hours or ~6 *S. aureus* generations in rich medium), are particularly concerning. In addition, the model that GpsB stimulates constriction of the FtsZ ring (Figure 6) is at odds with recent work from the Garner and Xiao laboratories indicating that FtsZ forms treadmilling polymers that serve to time average peptidoglycan synthesis but does not contribute directly to constriction itself (PMID:26831086 PMID: 28209899 PMID: 28209898). On the whole, I did not find the data sufficient to support the authors' major conclusions, significantly detracting from my initial enthusiasm for this study.

1) The value of ectopic GpsB expression in *B. subtilis* is limited at best. It is not uncommon for cell division proteins from one organism to interfere with division in another due to minor differences in protein-protein interaction. The fact that *S. aureus* gpsB does not complement a loss of function mutation in *B. subtilis* gpsB reinforces this point. GpsB depletion and overexpression in *S. aureus*, should be more than sufficient to prove their point provided they add additional information on FtsZ assembly in the different mutant strains.

2) In the absence of data indicating that excess GpsB and/or GpsB depletion disrupts FtsZ localization, I do not think that the authors can fairly conclude that GpsB inhibits FtsZ assembly in vivo. It is equally plausible that changes in GpsB impact the localization or activity of a downstream cell division protein.

3) Evidence for GpsB promoting lateral interactions between FtsZ polymers is limited to light scattering experiments that show increased signal when GpsB is added to FtsZ in vitro. Light scattering does not distinguish between differentiate between the formation of a few large protein complexes and the formation of a high number of smaller complexes (e.g. in the case of FtsZ large bundles or lots of single stranded polymers).

4) A better way to show GpsB driving lateral interaction is to add purified GpsB to FtsZ + GTP to determine if it a) increases pellet size in a sedimentation experiment and b) leads to the formation of bundles of single stranded polymers by EM.

5) The authors use the term polymerization when I believe they mean assembly. None of the assays they employ truly distinguish between polymerization-the formation of single stranded FtsZ polymers-and bundling-the formation of bundles of single stranded polymers via lateral interactions. Assembly is a generic term that the field uses to describe both phenomena.

6) I cannot find any information regarding GpsB protein concentration either in wild type cells or in overexpression strains. Such data is essential both for the reader to evaluate the impact of GpsB overexpression on *S. aureus* division and to support the physiological relevance of in vitro work.

7) For *E. coli* and *B. subtilis* FtsZ the leveling out in 90-degree angle assembly assays indicates that the reaction is at steady state with regard to polymerization and depolymerization. It does not mean that the polymers are "stable".

8) The concentration of Sa FtsZ used for in vitro experiments is quite high. In previous studies, Sa FtsZ activity was measured at 10μM (PMID: 22958099). The authors are encouraged to use FtsZ at as low a concentration as possible-ideally near its in vivo concentration-to avoid artifacts.

9) I was confused by the use of the His tag for purification of Sa FtsZ. Was this removed prior to analysis? If not, is the tagged version able to complement depletion of native Sa FtsZ in vivo? It is straightforward to purify native Sa FtsZ (see PMID 22958099), which is far preferable to the tagged version for in vitro analysis.

10) It is highly unlikely, if not physically impossible, for a protein that stimulates lateral interactions to enhance GTP hydrolysis. Lateral interactions stabilize FtsZ polymers, restricting nucleotide turnover and inhibiting hydrolysis.

11) The long (180 minute) delay between achievement of steady state and disassembly does not support the authors conclusion that Sa GpsB stimulates first bundle formation and then disassembly in vivo where FtsZ monomer turnover is likely on the order of seconds or minutes.

12) The time frames of the GTPase assays and the 90-degree angle light scattering experiments differ substantially. If the conditions of the two assays are the same and the authors' model is correct, the expectation is that GTP hydrolysis rates would be bimodal with a low rate early in the experiment (corresponding to high levels of light scattering and polymer bundling) and with a short burst of increased rates of GTP hydrolysis at the end (corresponding to disassembly).

13) The authors are encouraged to use a regenerating GTP system to avoid accumulation of GDP over long time courses. GDP inhibits FtsZ-FtsZ interactions and its accumulation could very well be the cause of the rapid disassembly observed in the LS assays.

14) The authors should consider alternative hypotheses. In particular, their data is consistent with a scenario in which GpsB stimulates FtsZ polymerization, elevating the concentration of short FtsZ polymers (but not bundles). Increases in the numbers of short polymers is expected to increase light scattering AND increase rates of GTP hydrolysis due to increases in the number of polymer ends available for nucleotide exchange.

Reviewer #3:

This report describes the GpsB protein of *Staphylococcus aureus*, which, like some other GpsB homologs, is involved cell division. Although GpsB is not essential for viability in *B. subtilis* or some strains of *S. pneumoniae*, it is essential in *S. aureus*. Here, the authors explore why GpsB might be essential. They first show that overproduction of *S. aureus* GpsB inhibits septation in *S. aureus* (causing cell enlargement) as well as in *B. subtilis* (causing cell filamentation), and nucleoid segregation seems to be affected in both cases. Depletion of GpsB in *S. aureus*, on the other hand, results in lysis of some cells and production of some anucleate cells, which is interpreted here as a cell division timing defect. The authors then show that GpsB localizes to the leading edge of the division septum, and purified GpsB, which seems to form multimers, co-sediments and co-assembles with FtsZ. *S. aureus* FtsZ has a very low GTPase activity and a high critical concentration, similar to *S. pneumoniae* FtsZ. However, when FtsZ is at high concentration, purified GpsB can stimulate the GTPase activity 3-fold and promote massive bundling of the highly concentrated FtsZ. Finally, light scattering experiments suggest that GpsB not only stimulates massive assembly of FtsZ bundles, but also eventually promotes its disassembly. An intriguing model proposes that GpsB first bundles FtsZ, then after a while debundles (or disassembles) FtsZ polymers, making it a dual-function FtsZ assembly state regulator.

Although the new information about *S. aureus* GpsB is interesting and welcome, particularly the direct interaction with FtsZ, this paper has a number of significant shortcomings. The main one is that the data do not provide sufficient support for many of the conclusions, including the regulation of assembly and disassembly of FtsZ described in the title. For example, key conclusions about FtsZ bundling and un-bundling are made from experiments with likely non-physiologically high levels of FtsZ (30 μM). The main conclusions about FtsZ bundling come from fluorescence experiments, but without inspection of these structures by EM, it is difficult to tell whether they are aggregates or bundles of protofilaments (especially as they are tens of microns in width, which would require thousands of FtsZ protofilaments to associate laterally).

Moreover, there is no differentiation between "assembly" and "bundling". As the relevant experiments in vitro use sedimentation, fluorescence microscopy of labeled proteins, and light scattering, all the observed effects may be different degrees of FtsZ protofilament bundling. It is too bad that the authors did not include some TEM analysis, as this could allow them to make more specific conclusions about assembly state, including GpsB assembly state. As it is, the fact that they don't really know whether the effects are on FtsZ lateral interactions or longitudinal interactions should at the least be stated in the text.

Another key conclusion, that GpsB promotes FtsZ disassembly, comes from a single experiment showing loss of light scattering from this very high concentration of FtsZ after incubation with GpsB. While the authors' conclusions may be correct, an alternative possibility is that the GTP was depleted. They mention use of a GTP regeneration system, but this system was not described or referenced, and there is no way to know whether such a regeneration system could be overwhelmed by 30 μm of assembled FtsZ that has been stimulated to hydrolyze GTP by GpsB.

Finally, there is a disconnect between the in vivo and in vitro results. For example, the in vivo images focus mainly on the effects of too much or too little GpsB on the nucleoid or membrane integrity, whereas in vitro experiments focus on GpsB-FtsZ interactions. It would greatly strengthen the authors' case if they could better correlate their in vivo and in vitro data. For example, there is no attempt to localize FtsZ in the GpsB depletion or overproducing strains.

Other major comments:

1) Figure 1E: the poorly segregated chromosomes in *B. subtilis* cells after GpsB overproduction are puzzling and were never discussed further.

2) Figure 1J-K: Why is the DAPI staining diffuse in panel J, while it is at the cell periphery in panel K? Why would DNA be at the cell periphery, unless GpsB itself drags DNA there? Is this a possibility? These odd results were never mentioned. The DAPI staining in Figures 2-3 looks much more normal.

3) Were the levels of GpsB overproduction similar in *B. subtilis* cells vs. *S. aureus* cells?

4) Subsection “Overproduction of *S. aureus* GpsB inhibits cell division in *B. subtilis* and *S. aureus*”, third paragraph and Figure 2: It seems that only a fraction of dividing *S. aureus* cells that are being depleted for GpsB have a nucleoid segregation defect. Unfortunately, the extent of this defect was not quantified, but from the images provided, only a few anucleate cells are apparent. Do the authors think that this defect is similar to the nucleoid segregation defect observed by Land et al. for *S. pneumoniae* in the absence of GpsB?

5) These experiments require much more rigor. First, the proportion of anucleate cells in the population needs to be quantitated. It is possible that to maintain overall ability to form a colony, a very high proportion of *S. aureus* cells must have nucleoids, but at this point it is impossible to tell. Second, the extent of GpsB depletion in the cell population (e.g., a western blot comparing protein levels during depletion with WT levels) was not shown, making it difficult to conclude much even if more quantitation of anucleate cells were done.

6) Subsection “Overproduction of *S. aureus* GpsB inhibits cell division in *B. subtilis* and *S. aureus*”, third paragraph: The accumulation of membrane debris was not at all apparent from the micrographs of FM4-64 staining, and neither were any explanations for this phenomenon or implications for the cell.

7) The time-lapse series in Figure 2D-F was not well presented. Although the cartoons to the right helped, it seems that the highlighted cells never formed visible division septa during the time course (contrary to what is claimed in the third paragraph of the subsection “Overproduction of *S. aureus* GpsB inhibits cell division in *B. subtilis* and *S. aureus*”), whereas septation in other cells, such as those at the right of the panel) seemed to progress in septation. Moreover, these cells had apparently normal segregated nucleoids. So it is not clear (a) how typical this time course is or (b) what proportion of the cells in this population have normal nucleoids and normal ability to segregate them.

8) Figure 2: The arrows and arrowheads were not defined. Also, there is no scale bar for panels D-F; they look slightly magnified compared with panels A-C.

9) Subsection “Overproduction of *S. aureus* GpsB inhibits cell division in *B. subtilis* and *S. aureus*”, end of third paragraph: the idea of a timing defect is speculative. In addition to the abnormal nucleoid distribution being a rare event (see comments #4-5 above), there is no direct evidence that there is a timing defect. Another possibility is that GpsB depletion somehow causes nucleoid segregation to run in reverse, at least in some cells.

10) Subsection “GpsB localizes to mid-cell and co-constricts with the divisome”, second paragraph: without doing a 3D reconstruction as was done in Figure 3Biii', it is not possible to confirm that Figure 3Bi shows a ring structure, although it is likely. Surely there must be some tilted cells that show the FtsZ/GpsB ring, as was shown in Pinho's Nature Reviews article.

11) Subsection “GpsB localizes to mid-cell and co-constricts with the divisome”, end of second paragraph: The localization of GpsB to the cell periphery in adjacent daughter cells instead of to new septal rings suggests that GpsB localizes later than FtsZ. This would be similar to what was observed for *S. pneumoniae* (Land et al.). It would have strengthened this section of the manuscript to show whether GpsB localizes later than FtsZ, and this would also test the model in Figure 6.

12) The high critical concentration for *S. aureus* FtsZ assembly suggests that this FtsZ needs a stimulatory factor for its assembly in vivo, and the massive increase in FtsZ bundling certainly makes a case for GpsB as a candidate for one of these factors. Localizing FtsZ in GpsB-depleted cells would be one way to validate this idea in vivo and provide some type of correlation between in vivo and in vitro results, for which there is none at present (see main comment above).

13) Subsection “GpsB stimulates GTPase activity of FtsZ”: Do the authors know what the approximate concentration of FtsZ is in *S. aureus* cells? FtsZ can form aberrant structures when it is highly concentrated, and these structures are also dependent on buffer and pH conditions. The authors have shown that *S. aureus* FtsZ's GTPase activity is heavily dependent on concentration, probably because its assembly is as well. Therefore, using concentrations of FtsZ that are several-fold higher than what is in cells may give artifactual results.

14)Subsection “GpsB stimulates GTPase activity of FtsZ”: The 3-fold stimulation of GTPase by GpsB is significant and interesting, but of course it is relative to a very low basal GTPase activity that, at a more physiological concentration of 10 μM, calculates to about 3 GTP hydrolyzed per FtsZ per hour. So that means GpsB can bring that number up to 10. The authors should discuss whether those rates of hydrolysis make physiological sense given the doubling time of *S. aureus* and the likelihood of FtsZ treadmilling at the septal ring (see two recent Science papers on this topic).

15) "did not appreciably stimulate" seems to be true, but perhaps the data suggest that GpsB might inhibit GTPase activity?

16) How reproducible is the separation of GpsB into hexamer and dodecamer fractions as shown in Figure 4C?

17) The y axes in Figure 4D-E are not the same as in 4C, where the rate per mol of FtsZ is a more intuitive way to present the data. Perhaps these could be harmonized.

18) The authors conclude that GpsB stimulates FtsZ GTPase activity sub-stoichiometrically. Does increasing GpsB concentration stimulate FtsZ's GTPase activity further?

19) What is the GpsB:FtsZ ratio in cells? How close to this ratio are the experimental conditions?

20) Subsection “GpsB interacts with and bundles FtsZ polymers”, second paragraph and Figure 5: Despite the claim in the text, no obvious structures are visible in the micrographs other than those in F-G.

21) Subsection “GpsB interacts with and bundles FtsZ polymers”, last paragraph and Figure 5N: it looks like FtsZ polymerized at the same rate in the presence of GpsB, just more extensively, so the wording should be changed. And as mentioned elsewhere, "polymerized" likely reflects bundling of protofilaments, not polymerization of single protofilaments.

22) How reproducible is the disappearance of fluorescent FtsZ polymers and loss of light scattering signal in response to GpsB? This is certainly an intriguing result, but needs more rigorous corroboration before it can be a major conclusion of the paper (and in the title). Even then, any conclusion about GpsB regulation of FtsZ assembly needs to come with the caveat that GpsB likely interacts with multiple divisome proteins (e.g. see Rued et al., 2017, for examples in *S. pneumoniae*) that may modulate its activities.

23) Discussion, first paragraph: As there is no in vivo evidence to back up the in vitro data, this conclusion seems premature.

24) Discussion, second paragraph: the logic here is cloudy, as it is not clear why depletion of a protein that both positively and negatively regulates FtsZ assembly could result in incorrect timing of cell division, particularly as this incorrect timing was not directly shown (see comments above).

25) "GpsB rapidly colocalized with the cell division machinery at the onset of cytokinesis". This is partially true, but perhaps more importantly, there seems to be a delay between the arrival of FtsZ and that of GpsB, because GpsB is not localized to the ring (and yet is localized to the membrane) in newborn cells. If GpsB is important for FtsZ ring assembly, which is implied by the model, how might that be reconciled?

26) "consistent with its role in modulating the activity of FtsZ". It is also consistent with its ability to bind a divisome component, which could be FtsZ but also could be something else such as EzrA.

27) Discussion, third paragraph: In their model, the authors assume that FtsZ disassembly is a feature of ring constriction. While this may be true, there is also published evidence that increased FtsZ filament density is important for ring constriction. At this point, it is totally unclear what happens to FtsZ assembly state during constriction in *S. aureus*.

28) Discussion, third paragraph and following are indeed very speculative considering that they are largely based on one gel filtration experiment.

29) Discussion, third paragraph: Might another model be that the L35S mutant GpsB fails to redistribute in the cell because it interacts poorly with FtsZ?

30) Discussion, third paragraph: While the correlation between EzrA and GpsB may be true, it is misleading to correlate them because they both have similar depletion phenotypes; depletion of any protein important for *S. aureus* cell division will result in cell enlargement.

31) Discussion, last paragraph: If this model is true, then an interesting and straightforward experiment would be to determine if there is a time lag in GTP hydrolysis compared with assembly. The prediction is that assembly should occur prior to hydrolysis.

32) Figure 6: although quite a bit of text early in the paper is devoted to the aberrant nucleoid segregation in response to excess or depleted GpsB levels, the nucleoid is ignored in the model.

33) Figure 6: the model for GpsB is reminiscent of a model for ZipA and FtsA in *E. coli*, with ZipA as the FtsZ bundler and FtsA as the debundler.

34) Do the authors know whether the His_6_ tags on both of the proteins have any aberrant effect on their interactions?

[Editors’ note: what now follows is the decision letter after the authors submitted for further consideration.]

Thank you for submitting your article "An essential *Staphylococcus aureus* cell division protein directly regulates FtsZ dynamics" for consideration by *eLife*. Your article has been reviewed by three peer reviewers, and the evaluation has been overseen by a Reviewing Editor and Anna Akhmanova as the Senior Editor. The following individual involved in review of your submission has agreed to reveal his identity: William Margolin (Reviewer #2).

The reviewers have discussed the reviews with one another and the Reviewing Editor has drafted this decision to help you prepare a revised submission.

Summary:

This revised manuscript represents a significant improvement over the original submission. Most importantly, biochemical and cytological experiments are for the most part now sufficiently rigorous and multiple controls have been added to the benefit of the entire study.

Essential revisions:

It is proposed that higher FtsZ bundling and thus higher local concentrations lead to higher GTPase activity. As mentioned in the Discussion this would be in contrast to other proteins such as *E. coli* ZapA or ZapC that bundle FtsZ without activating GTPase activity. The sedimentation results show nicely that GpsB interacts with FtsZ; but if GpsB bundles FtsZ, why then would its addition not increase the amount of FtsZ in the pellet fraction? Is the fraction of FtsZ bundles already maximized in these conditions? This should be commented upon because the sedimentation results are currently at odds with the light scattering experiments.

An alternative model consistent with the sedimentation data that the authors might consider is that GTP hydrolysis is not a consequence of FtsZ bundling but instead is required for bundling, perhaps because GpsB-mediated FtsZ bundling only occurs after FtsZ subunits become poised for turnover. In other words, GpsB stimulates GTP hydrolysis by FtsZ, which then triggers a conformational change that stimulates lateral interactions between FtsZ protofilaments. This could be tested by repeating the sedimentation with GTP instead of GMPCPP, matching the conditions in the light scattering experiment. If enough GTP is provided, e.g. 4 mM or so, the low hydrolysis rate of FtsZ makes it unlikely that GTP will be exhausted during the experiment. The prediction is that the ability of FtsZ to hydrolyze GTP will allow GpsB to bundle FtsZ and consequently pellet much more efficiently. This also predicts that FtsZ bundles would not be detectable by EM in the presence of GpsB and GMPCPP instead of GTP.

---

## [Author Response]

[Editors’ note: the author responses to the first round of peer review follow.]

Reviewer #1:In this manuscript, the authors analyze the function of the GpsB protein in S. aureus. GpsB and its paralog DivIVA are well-studied in Firmicutes and Actinobacteria. In these species, these two proteins have been shown to be involved in regulation of cell growth, regulation of cell division and/or in chromosome segregation. Here, it is reported that overproduction of GpsB from S. aureus in B. subtilis gives rise to filamentous cells; overproduction of GpsB in S. aureus gives rise to large spherical cells; depletion of GpsB in S. aureus results in inhibition of cell division; a gspB mutation is isolated that does not affect cell shape upon expression in B. subtilis; GpsB localization is determined and found to be dynamic over the cell cycle: and, that GpsB interacts with FtsZ in vitro.Previously, it has been reported that GpsB is essential in S. aureus. This manuscript aims to clarify the function of GpsB in S. aureus. The data presented present a good starting point for this clarification; however, as outlined below I have several major problems with the manuscript that will require additional experiments. In its current format, the data do not support the major conclusions.

We are pleased that the reviewer appreciated that this was a good starting point to clarify the function of GpsB. We have responded to all the reviewer’s concerns below, with fresh experiments when required, to ensure that this becomes a more complete story.

1) In the Introduction it would be very helpful if an explanation was included for how DivIVA and GpsB proteins are defined and how they differ from each other.

Agreed. We have now included a comparison of the DivIVA and GpsB structural features in the Introduction (second paragraph).

GpsB expression in B. subtilis:It is not clear that GpsB from S. aureus and B. subtilis accumulate at the same levels in the strains were the two proteins are overproduced.

We have now included an immunoblot (new Figure 1—figure supplement 1B) that indicates the extent of GpsB^Sa^ overproduction in *B. subtilis* in the presence and absence of inducer (note: the GpsB^Sa^ antibody does not recognize the *B. subtilis* GpsB ortholog). Additionally, we have included an immunoblot that indicates the extent of overproduction of plasmid‐encoded GpsB^Sa^ in *S. aureus* (new Figure 1—figure supplement 1C). The data are discussed in the first and third paragraphs of the subsection “Overproduction of *S. aureus* GpsB inhibits cell division in *B. subtilis* and *S. aureus*”.

The multiple nucleoids in the filamentous B. subtilis cells formed after overproduction of GpsB from S. aureus are not clearly segregated. Therefore, it is not clear what actually causes the filamentous phenotype, i.e. does GpsB from S. aureus interfere with chromosome segregation or with cell division or both processes?

We thank the reviewer for pointing this out. We re‐imaged these strains, paying more attention to the chromosomal arrangement, and found that the nucleoids are indeed well segregated in the filamentous strains, suggesting that the filamentation defect in *B. subtilis* is independent of chromosome segregation. The revised image is now reported in a new Figure 1B‐M. This nucleoid segregation pattern was also evident in the various *B. subtilis* deletion strains we tested (please see next response). We have therefore now stated more explicitly that GpsB specifically interferes with cell division.

Does overproduction of GpsB from S. aureus cause a filamentous phenotype in a B. subtilis strain that lacks the native GpsB or DivIVA?

We appreciate the suggested experiment and took this a step further. In addition to *B. subtilis* GpsB and DivIVA, we tested the dependence of every reported GpsB‐interacting protein (PrkC, PonA, and EzrA) for the filamentation phenotype. The results, presented in a new Figure 1B‐M, indicated that *B. subtilis* GpsB, DivIVA, PrkC, PonA, and EzrA are *not* required for GpsB^Sa^‐mediated *B. subtilis* filamentation.

GpsB overproduction in S. aureus:Please provide numbers for how many uninduced cells have septa and how many cells have septa after induction of GpsB overproduction.

We have now reported the percentage of cells <1.2 μm in diameter and >1.2 μm in diameter in WT, WT + vector, and upon overproduction of the *S. aureus* and *B. subtilis* ortholog of GpsB. The microscopy data are shown in a new Figure 2 and the quantification are reported in the third paragraph of the subsection “Overproduction of *S. aureus* GpsB inhibits cell division in *B. subtilis* and *S. aureus*”.

The large cells formed after GpsB overproduction seem to be devoid of nucleoid. So more quantitative analyses using chromosomal markers need to be included to clearly demonstrate that chromosome segregation is normal.From the data provided it is not clear if overproduced GpsB interferes with chromosome segregation, cell division, cell growth or a combination of these processes. It is essential to include additional experiments to precisely define the process that overproduced GpsB interferes with is precisely defined.

In the revised version, since we have shown that overproduction of GpsB^Sa^ in *B. subtilis* indeed did not result in a defect in chromosome segregation, we have removed the DAPI images in the revised version of the figure (now Figure 2A‐J). Instead, we have more carefully shown that overproduction of the *B. subtilis* ortholog or the L35S variant does not result in cell enlargement.

GpsB depletion in S. aureus:Please include experimental data to demonstrate that GpsB has been depleted in the experiments shown in Figure 2.

We have now included an immunoblot (new Figure 1—figure supplement 1F) that shows the depletion of GpsB upon induction of the antisense. It is worth noting that induction of the antisense RNA resulted in cell lysis, which prevented efficient extraction of bulk protein from the culture. We therefore induced the construct and collected samples at an earlier time point, before cell lysis took over the culture, and quantified the fold decrease. As a result, the GpsB that we detected is likely over‐represented and the fold‐depletion we calculated is likely an underestimation.

Along the same lines, it seems that a construct in which expression of gpsB is regulated directly (and without the extra step with the antisense RNA) would be a more simple experimental setup.

We agree! This was certainly not our preferred method to test the depletion phenotype of GpsB. However, we were unsuccessful (multiple times) in knocking out the native copy of *gpsB* in the presence of an inducible copy of the *gpsB*, which is why we attempted the antisense RNA method for knocking down *gpsB*. We would point out that a similar antisense depletion strategy was recently reported by Pinho’s group for the depletion of DivIB, DivIC, and FtsL in *S. aureus* (Nature, 2018 PMID: 29443967).

Only a few S. aureus cells seem to be affected in their cell shape in the depletion experiment shown in Figure 2A, B. More quantitative data are needed.

Agreed. We have now repeated the time lapse depletion experiment and show a field in which every cell shows a cell division defect (new Figure 2L‐M). We report that depletion of GpsB prevents initiation of cell division in cells that have not started dividing and arrests cell division in cells that have initiated cell division. Cells also accumulate aberrant membrane accumulations that preceded lysis.

From the experiments in Figure 2B it is by no means clear that chromosome segregation proceeds normally. In fact, it looks as if segregation has been halted. So, more quantitative analyses using chromosomal markers need to be included to clearly demonstrated that chromosome segregation is normal.As is the case for the overproduction experiment, it is not clear if the depletion phenotype is caused by defects in chromosome segregation, cell division, cell growth or a combination of these processes. It is essential to include additional experiments to precisely define the process that is affected upon GpsB depletion.

In the revised version of the paper, we have now shown the direct involvement of GpsB in cell division by examining its effect on the divisome directly (ZapA, a proxy for FtsZ) and, conversely by demonstrating the dependence of GpsB localization on the divisome. Moreover, we have shown in a purified system that GpsB is sufficient to influence FtsZ assembly kinetics. Taken together, we hope that we have convinced the reviewer that the phenotypes can be explained by a direct interaction of GpsB with the divisome.

GpsB^L35S^:What is the phenotype of the gpsB^L35S^ mutant in S. aureus?

We certainly sought an answer to this question ourselves, but ultimately this is the only requested experiment that we were unable to perform exactly as requested, entirely due to technical reasons. Since we were unable to obtain a clean *gpsB* deletion strain (despite our best efforts), we were unable to complement it with an inducible copy of WT *gpsB* or *gpsB^L35S^*. We are currently attempting other strategies to perform this experiment, but did not want to delay submission of the current manuscript.

That said, we did test the effect of the overproduced L35S allele in a merodiploid strain of *S. aureus* that also harbored the WT copy of *gpsB*. The results indicate that the L35S allele is recessive in vivo in *S. aureus*. We have included this as a revised Figure 2I‐J. We hope that our identification of this allele, the demonstration that it loses function when expressed in *B. subtilis*, is recessive in vivo in *S. aureus*, and the demonstration that it does not function in vitro will be sufficient in this report.

Cell biology:The experiments nicely demonstrate that GpsB localize to the septum. Does this localization depend on FtsZ, i.e. what happens to GpsB localization upon depletion of FtsZ?

We have now monitored localization of GpsB in cells treated with the FtsZ inhibitor PC190723. A similar strategy was recently reported by Pinho’s group for studying the effects of FtsZ depletion in *S. aureus* (Nature, 2018 PMID: 29443967). The results, presented in a new Figure 4A‐F’, indicate that GpsB localization depends on FtsZ.

To establish a more direct connection between cell division and GpsB it is essential to include experiments on Z-ring formation and localization in the strains overproducing GpsB or depleted for GpsB.

Agreed. We examined the localization of ZapA‐GFP, which has been used as a proxy for FtsZ localization. Upon depletion of GpsB, the cells eventually lysed. However, before they lysed, we examined ZapA‐GFP localization and found that Z‐rings formed at mid‐cell as GpsB was depleted, but the intensity of the ZapA‐ring was ~4‐fold fainter than WT. We concluded that GpsB promotes the robust assembly of the Z‐ring and that FtsZ and GpsB reciprocally influence each other. The results are presented in a new Figure 4J‐K’. In cells overproducing GpsB, ZapA‐GFP mislocalized in 86% of the enlarged cells. These results are shown in Figure 4G‐I’.

in vitro experiments:Please use the same units in Figure C, D and E.

GTP hydrolysis figures referenced now express turnover in pmol min^‐1^ or pmol min^‐1^ pmol FtsZ^‐1^, as appropriate.

All experiments in Figure 5: Please include EM images to unequivocally demonstrate that FtsZ forms filaments and that these filaments are altered by GpsB.

We have now replaced the in vitro fluorescence micrographs in (now) Figure 6 with purified FtsZ and GpsB with EM images. The images are qualitatively similar to those published by the Erickson lab (see Author response image 1; Panels A-C (left) are reproduced from Figure 4 of Chen et al. 2017, Scientific Reports 7:3650, published under the terms of the Creative Commons Attribution 4.0 International license (CC BY 4.0; http://creativecommons.org/licenses/by/4.0/))) that demonstrated the ability of *E. coli* ZipA to bundle FtsZ filaments (please compare arrowheads to arrowheads; arrows to arrows).

**Author response image 1. respfig1:** Bundling of *E. coli* FtsZ filaments by ZipA, compared to bundling of *S. aureus* FtsZ filaments by GpsB.

All experiments in Figure 5: Please include control experiments that the effect of GTP on FtsZ polymerization is specific to GTP.

We have tested *S. aureus* FtsZ polymerization in the presence of GTP, GDP, ATP, and ADP. The data (included as a new Figure 6—figure supplement 1A) indicate that, like orthologs of FtsZ from other bacteria, *S. aureus* FtsZ only polymerizes in the presence of GTP.

Reviewer #2:Eswara et al. describe analysis of the S. aureus homolog of GpsB, a protein implicated in varied aspects of cell division among the firmicutes. Using several approaches they come to the conclusion that in contrast to other bacteria, Sa GpsB interacts directly with the highly conserved cell division protein FtsZ to control its activity. I was initially enthusiastic about this study and its potential to define the function of GpsB, a protein whose true nature has been the source of significant debate in the field. Unfortunately, many of the experiments presented in this paper are poorly designed and the physiological relevance of the authors' findings is unclear. Experiments investigating the impact of GpsB on B. subtilis growth and division, the apparent inclusion of an N-terminal His tag on Sa FtsZ in sedimentation and light scattering experiments, and the extremely long duration of the light scattering assays (3 hours or ~6 S. aureus generations in rich medium), are particularly concerning. In addition, the model that GpsB stimulates constriction of the FtsZ ring (Figure 6) is at odds with recent work from the Garner and Xiao laboratories indicating that FtsZ forms treadmilling polymers that serve to time average peptidoglycan synthesis but does not contribute directly to constriction itself (PMID:26831086 PMID: 28209899 PMID: 28209898). On the whole, I did not find the data sufficient to support the authors' major conclusions, significantly detracting from my initial enthusiasm for this study.

In the revised version, we have responded to every concern raised, often with fresh experiments. Notably, all the in vitro experiments have been repeated with an untagged *S. aureus* FtsZ, and we now explicitly discuss in the text how these results may be consistent with a treadmilling activity for FtsZ. The result, we think, is a greatly improved manuscript that will we hope will reignite the reviewer’s initial enthusiasm.

1) The value of ectopic GpsB expression in B. subtilis is limited at best. It is not uncommon for cell division proteins from one organism to interfere with division in another due to minor differences in protein-protein interaction. The fact that S. aureus gpsB does not complement a loss of function mutation in B. subtilis gpsB reinforces this point. GpsB depletion and overexpression in S. aureus, should be more than sufficient to prove their point provided they add additional information on FtsZ assembly in the different mutant strains.

We agree that the *S. aureus* experiments are ultimately the most informative, but the transcomplementation experiments in *B. subtilis* foreshadow the sufficiency of GpsB in interacting with FtsZ that is formally demonstrated in the biochemical experiments at the end of the paper. Moreover, reviewers 1 and 3 asked for additional experiments in *B. subtilis* to clarify our initial observation. In the revised version, we have therefore reported these experiments (and the experiments requested by the other reviewers) in a new Figure 1, but we are of course open to removing all the *B. subtilis* data if the reviewers suggest it is appropriate.

*2) In the absence of data indicating that excess GpsB and/or GpsB depletion disrupts FtsZ localization, I do not think that the authors can fairly conclude that GpsB inhibits FtsZ* assembly *in vivo. It is equally plausible that changes in GpsB impact the localization or activity of a downstream cell division protein.*

Agreed. We have examined the localization of FtsZ‐GFP in *B. subtilis* when GpsB was overexpressed and observed that FtsZ‐GFP itself mis‐localized in the cytosol. The data are now reported in Figure 1N‐O.

3) Evidence for GpsB promoting lateral interactions between FtsZ polymers is limited to light scattering experiments that show increased signal when GpsB is added to FtsZ in vitro. Light scattering does not distinguish between differentiate between the formation of a few large protein complexes and the formation of a high number of smaller complexes (e.g. in the case of FtsZ large bundles or lots of single stranded polymers).

We have now directly examined lateral interactions between FtsZ polymers using EM. Please see our detailed response to comment #4 below.

4) A better way to show GpsB driving lateral interaction is to add purified GpsB to FtsZ + GTP to determine if it a) increases pellet size in a sedimentation experiment and b) leads to the formation of bundles of single stranded polymers by EM.

We had included a sedimentation experiment in the original submission that has now been repeated with untagged FtsZ in the current submission. In the original submission, we also showed lateral interaction in vitro with fluorescently FtsZ polymers. In the present submission, we have replaced this experiment in Figure 6 with EM images that more directly demonstrate bundling of FtsZ polymers. As we noted in the response to reviewer 1 (Author response image 1, the images are qualitatively similar to those published by the Erickson lab (Figure 4A-C in Sci Rep 2017; 7:3650) that demonstrated the ability of *E. coli* ZipA to bundle FtsZ filaments (please compare arrowheads to arrowheads; arrows to arrows).

5) The authors use the term polymerization when I believe they mean assembly. None of the assays they employ truly distinguish between polymerization-the formation of single stranded FtsZ polymers-and bundling-the formation of bundles of single stranded polymers via lateral interactions. Assembly is a generic term that the field uses to describe both phenomena.

Agreed. We have substituted “polymerization” with “assembly” to describe FtsZ behavior in the light scattering and centrifugation experiments. We use the term “bundling” when we directly observe it in the new EM experiments.

6) I cannot find any information regarding GpsB protein concentration either in wild type cells or in overexpression strains. Such data is essential both for the reader to evaluate the impact of GpsB overexpression on S. aureus division and to support the physiological relevance of in vitro work.

We have now included a discussion comparing our in vitro data to the physiological concentration of FtsZ and GpsB in *S. aureus* (Discussion, first paragraph). Briefly, FtsZ and GpsB are reported to be present at 4452 and 1659 molecules per cell (Zuhlke et al., 2016; primary data found in http://aureowiki.med.unigreifswald.de/SACOL1199 and http://aureowiki.med.uni‐greifswald.de/SACOL1484), which is equivalent to 7.4 × 10^‐21^ moles of FtsZ. If we consider a *S. aureus* cell with a diameter of 0.8 µm, the cell volume is 2.7 × 10^‐16^L, so the molarity of FtsZ in vivo is 27.6 µM: remarkably close to the 30 µM that was required for our purified FtsZ to show appreciable activity. The intracellular concentration of GpsB is 2.7‐fold less than FtsZ, or 10.2 µM‐ again, remarkably close to the 10 µM we used in our in vitro experiments.

We have now included, in a new Figure 1—figure supplement 1B‐C, immunoblots that show the extent of overproduction of GpsB in *B. subtilis* and *S. aureus*.

7) For E. coli and B. subtilis FtsZ the leveling out in 90-degree angle assembly assays indicates that the reaction is at steady state with regard to polymerization and depolymerization. It does not mean that the polymers are "stable".

We agree with the reviewer that this was a very poor choice of words. We have now described the plateau as indicating that the reaction is in steady state.

8) The concentration of Sa FtsZ used for in vitro experiments is quite high. In previous studies, Sa FtsZ activity was measured at 10μM (PMID: 22958099). The authors are encouraged to use FtsZ at as low a concentration as possible-ideally near its in vivo concentration-to avoid artifacts.

Please see the detailed response to point #6. The intracellular concentration of FtsZ in *S. aureus* is ~28 µM and that of GpsB is ~10 µM‐ remarkably similar to the concentrations required in our in vitro experiments that showed activity. Thus, we have used FtsZ and GpsB at concentrations that are nearly identical to their in vivo concentrations.

The cited paper reported that “…the basal activity of the enzyme is significantly lower (∼0.5 GTP/FtsZ/min) than we observed for EcFtsZ or BsFtsZ.” By comparison, we too observed FtsZ activity (~0.1 GTP/FtsZ/min; Figure 5D) when we used 10 µM FtsZ‐ a very similar turnover as reported by Anderson et al. We have now cited this paper in the Results section as consistent with the slow GTP turnover rate by FtsZ reported here. Additionally, we now have included experiments in Figure 5F that test the effect of GpsB on 10 µM FtsZ. There is indeed a slight but insignificant increase in GTPase activity, but the overall rates are so low that it is difficult to draw a firm conclusion.

9) I was confused by the use of the His tag for purification of Sa FtsZ. Was this removed prior to analysis? If not, is the tagged version able to complement depletion of native Sa FtsZ in vivo? It is straightforward to purify native Sa FtsZ (see PMID 22958099), which is far preferable to the tagged version for in vitro analysis.

We have now replaced all previous experiments in the manuscript with experiments that use only untagged *S. aureus* FtsZ. The data are now reported in new Figures 5‐6.

10) It is highly unlikely, if not physically impossible, for a protein that stimulates lateral interactions to enhance GTP hydrolysis. Lateral interactions stabilize FtsZ polymers, restricting nucleotide turnover and inhibiting hydrolysis.

We certainly appreciate that these two seemingly contrasting activities have not been observed previously‐ indeed, we ourselves initially struggled with reconciling these observations. That said, we have demonstrated, in a purified system, that 1) addition of GpsB alone stimulates GTP hydrolysis of FtsZ, and 2) promotes lateral interactions between FtsZ polymers as measured by electron microscopy. The model we have proposed states that promoting lateral interactions between FtsZ filaments (which we refer to as “bundling”, in keeping with the nomenclature for similar in vitro phenomena observed using *E. coli* for example) increases the local concentration of FtsZ and increases GTPase activity. A slight variation of this model, which we also included in the Discussion, is that GpsB may remodel FtsZ conformation to “stimulate” GTPase activity.

It is critical to note that, for previously reported proteins, FtsZ bundling activity was the endpoint of their activity. Here, we are proposing that bundling is an intermediate step that allows for the stimulation of GTPase activity.

11) The long (180 minute) delay between achievement of steady state and disassembly does not support the authors conclusion that Sa GpsB stimulates first bundle formation and then disassembly in vivo where FtsZ monomer turnover is likely on the order of seconds or minutes.

Agreed. We have revised the model to suggest that enhanced GTP turnover by FtsZ, stimulated by GpsB, likely leads to increased treadmilling, which drives progression through the cell cycle. Light scattering assays performed on FtsZ, after removal of the histidine tag and shown in Figure 6, suggest that steady state is reached in ~10‐25 min and shortly thereafter, disassembly is favored. In the in vitro assay, this is due to GTP depletion.

12) The time frames of the GTPase assays and the 90-degree angle light scattering experiments differ substantially. If the conditions of the two assays are the same and the authors' model is correct, the expectation is that GTP hydrolysis rates would be bimodal with a low rate early in the experiment (corresponding to high levels of light scattering and polymer bundling) and with a short burst of increased rates of GTP hydrolysis at the end (corresponding to disassembly).

As described in the response to point #11, we have revised the model to suggest that enhanced GTP turnover by FtsZ, stimulated by GpsB, likely leads to increased treadmilling, which drives progression through the cell cycle. We did not detect either a lag or bimodal rates in time course experiments following GTP hydrolysis. Light scattering assays performed on FtsZ, after removal of the histidine tag and shown in Figure 6, are shorter (signal begins to fall by ~25 min) and consistent with GTP hydrolysis assays (i.e., 60 min or less).

13) The authors are encouraged to use a regenerating GTP system to avoid accumulation of GDP over long time courses. GDP inhibits FtsZ-FtsZ interactions and its accumulation could very well be the cause of the rapid disassembly observed in the LS assays.

Agreed. We now report (in a new Figure 6—figure supplement 1B) that, in the presence of a regeneration system, FtsZ does not disassemble during this time period. The argument presented in Figure 6L, where FtsZ disassembles after “hyper” assembly in the presence of GpsB, is that the assembly and bundling is reversible. We suggest that the enhanced GTP turnover may correlate to the onset of rapid treadmilling activity in vivo.

14) The authors should consider alternative hypotheses. In particular, their data is consistent with a scenario in which GpsB stimulates FtsZ polymerization, elevating the concentration of short FtsZ polymers (but not bundles). Increases in the numbers of short polymers is expected to increase light scattering AND increase rates of GTP hydrolysis due to increases in the number of polymer ends available for nucleotide exchange.

Actually, we think that the reviewer’s model is quite consistent with what we have proposed. In the new model figure (Figure 7B), we purposely show discontinuous FtsZ and GpsB localization at the division site. Our model now explicitly proposes that GpsB increases local concentration of FtsZ, which triggers GTPase activity, which may be linked to the onset of treadmilling.

Reviewer #3:[…] Although the new information about S. aureus GpsB is interesting and welcome, particularly the direct interaction with FtsZ, this paper has a number of significant shortcomings. The main one is that the data do not provide sufficient support for many of the conclusions, including the regulation of assembly and disassembly of FtsZ described in the title. For example, key conclusions about FtsZ bundling and un-bundling are made from experiments with likely non-physiologically high levels of FtsZ (30 μM). The main conclusions about FtsZ bundling come from fluorescence experiments, but without inspection of these structures by EM, it is difficult to tell whether they are aggregates or bundles of protofilaments (especially as they are tens of microns in width, which would require thousands of FtsZ protofilaments to associate laterally).Moreover, there is no differentiation between "assembly" and "bundling". As the relevant experiments in vitro use sedimentation, fluorescence microscopy of labeled proteins, and light scattering, all the observed effects may be different degrees of FtsZ protofilament bundling. It is too bad that the authors did not include some TEM analysis, as this could allow them to make more specific conclusions about assembly state, including GpsB assembly state. As it is, the fact that they don't really know whether the effects are on FtsZ lateral interactions or longitudinal interactions should at the least be stated in the text.Another key conclusion, that GpsB promotes FtsZ disassembly, comes from a single experiment showing loss of light scattering from this very high concentration of FtsZ after incubation with GpsB. While the authors' conclusions may be correct, an alternative possibility is that the GTP was depleted. They mention use of a GTP regeneration system, but this system was not described or referenced, and there is no way to know whether such a regeneration system could be overwhelmed by 30 μm of assembled FtsZ that has been stimulated to hydrolyze GTP by GpsB.Finally, there is a disconnect between the in vivo and in vitro results. For example, the in vivo images focus mainly on the effects of too much or too little GpsB on the nucleoid or membrane integrity, whereas in vitro experiments focus on GpsB-FtsZ interactions. It would greatly strengthen the authors' case if they could better correlate their in vivo and in vitro data. For example, there is no attempt to localize FtsZ in the GpsB depletion or overproducing strains.

We are glad to read that the reviewer found our paper interesting and welcome! In the revised version, we have responded to each of the reviewer’s concerns with fresh experiments as required, including examining the bundled FtsZ structures using EM. Specific responses to each major point is included below.

Other major comments:1) Figure 1E: the poorly segregated chromosomes in B. subtilis cells after GpsB overproduction are puzzling and were never discussed further.

We re‐imaged these strains paying more attention to the chromosomal arrangement, and found that the nucleoids are indeed well segregated in the filamentous strains (WT and various mutant backgrounds requested by reviewer 1). We have now omitted these statements.

2) Figure 1J-K: Why is the DAPI staining diffuse in panel J, while it is at the cell periphery in panel K? Why would DNA be at the cell periphery, unless GpsB itself drags DNA there? Is this a possibility? These odd results were never mentioned. The DAPI staining in Figures 2-3 looks much more normal.

The DAPI localization at the periphery was puzzling to us as well, but is likely well beyond the scope of the manuscript, which focuses on the direct interaction between GpsB and FtsZ. So that this observation can receive the proper amount of analysis, in the revised version, we have omitted the image and will follow up on this phenomenon elsewhere where we can dedicate a sufficient amount of space to analyzing the results.

3) Were the levels of GpsB overproduction similar in B. subtilis cells vs. S. aureus cells?

We have now included, in a new Figure 1—figure supplement 1B‐C, immunoblots that show the bulk overproduction levels of GpsB in *B. subtilis* and *S. aureus* and have reported the approximate fold‐overproduction in the main text (subsection “Overproduction of *S. aureus* GpsB inhibits cell division in *B. subtilis* and *S. aureus*”, first and third paragraphs). Overproduction was between ~3.5‐5 ‐fold.

4) Subsection “Overproduction of S. aureus GpsB inhibits cell division in B. subtilis and S. aureus”, third paragraph and Figure 2: It seems that only a fraction of dividing S. aureus cells that are being depleted for GpsB have a nucleoid segregation defect. Unfortunately, the extent of this defect was not quantified, but from the images provided, only a few anucleate cells are apparent. Do the authors think that this defect is similar to the nucleoid segregation defect observed by Land et al. for S. pneumoniae in the absence of GpsB?

Agreed. The depletion experiment was initially a bit challenging for us, but after optimizing conditions, we have now repeated the time lapse depletion experiment and show a field in which every cell shows a cell division defect (new Figure 2L‐M). We report that depletion of GpsB prevents initiation of cell division in cells that have not started dividing and arrests cell division in cells that have initiated cell division. Cells also accumulate aberrant membrane accumulations that preceded lysis. Due to phototoxicity issues, we did not monitor chromosome segregation. We hope that, for the current report showing the direct interaction between FtsZ and GpsB, that we may focus on the cell division defects caused by depletion of GpsB.

5) These experiments require much more rigor. First, the proportion of anucleate cells in the population needs to be quantitated. It is possible that to maintain overall ability to form a colony, a very high proportion of S. aureus cells must have nucleoids, but at this point it is impossible to tell. Second, the extent of GpsB depletion in the cell population (e.g., a western blot comparing protein levels during depletion with WT levels) was not shown, making it difficult to conclude much even if more quantitation of anucleate cells were done.

We have now included an immunoblot (new Figure 1—figure supplement 1F) that shows the depletion of GpsB upon induction of the antisense RNA. It is worth noting that induction of the antisense RNA resulted in cell lysis, which prevented efficient extraction of bulk protein from the culture. We therefore induced the construct and collected samples at an earlier time point, before cell lysis took over the culture, and quantified the fold decrease. As a result, the GpsB that we detected is likely over‐represented and the fold‐depletion we calculated is likely an underestimation. The new experiment now combines a larger field of cells that display the cell division phenotype, quantification of cells that show cell division defects, and a clear demonstration that GpsB is being depleted.

6) Subsection “Overproduction of S. aureus GpsB inhibits cell division in B. subtilis and S. aureus”, third paragraph: The accumulation of membrane debris was not at all apparent from the micrographs of FM4-64 staining, and neither were any explanations for this phenomenon or implications for the cell.

In a revised Figure 2L‐M, we have now included time lapse images of cells that are being depleted of GpsB. In the image, we show the aberrant accumulation of membrane upon GpsB depletion and point out two populations of cells: those that have initiated cell division, but become arrested; and those that have not initiated cell division and fail to initiate. Both cell types accumulate aberrant membranes and ultimately lyse. We interpret membrane accumulation as likely resulting from impaired cell division.

7) The time-lapse series in Figure 2D-F was not well presented. Although the cartoons to the right helped, it seems that the highlighted cells never formed visible division septa during the time course (contrary to what is claimed in the third paragraph of the subsection “Overproduction of S. aureus GpsB inhibits cell division in B. subtilis and S. aureus”), whereas septation in other cells, such as those at the right of the panel) seemed to progress in septation. Moreover, these cells had apparently normal segregated nucleoids. So it is not clear (a) how typical this time course is or (b) what proportion of the cells in this population have normal nucleoids and normal ability to segregate them.

We have now optimized the conditions for growth under the agarose pad and the imaging to reduce phototoxicity. The result, we think, is a much clearer representation of the cell division arrest phenotype, so we have eschewed the cartoon representation for this figure (now Figure 2L‐M).

8) Figure 2: The arrows and arrowheads were not defined. Also, there is no scale bar for panels D-F; they look slightly magnified compared with panels A-C.

We thank the reviewer for catching this. In the revised Figure 2, all arrows and arrowheads are now defined in the legend, and scale bars are included for images represented at different magnifications.

9) Subsection “Overproduction of S. aureus GpsB inhibits cell division in B. subtilis and S. aureus”, end of third paragraph: the idea of a timing defect is speculative. In addition to the abnormal nucleoid distribution being a rare event (see comments #4-5 above), there is no direct evidence that there is a timing defect. Another possibility is that GpsB depletion somehow causes nucleoid segregation to run in reverse, at least in some cells.

We agree. We have now softened the conclusion here and claim only that 1) GpsB depletion arrests cell division, 2) causes the aberrant accumulation of membranes, and 3) ultimately results in cell lysis.

10) Subsection “GpsB localizes to mid-cell and co-constricts with the divisome”, second paragraph: without doing a 3D reconstruction as was done in Figure 3Biii', it is not possible to confirm that Figure 3Bi shows a ring structure, although it is likely. Surely there must be some tilted cells that show the FtsZ/GpsB ring, as was shown in Pinho's Nature Reviews article.

We have now included 3D reconstructions of all 2D images that were rotated around the y‐axis to check if GpsB formed ring‐like structures. The results are presented in a new Figure 3B, which show that GpsB forms a single ring structure at the onset of cell division, co‐constricts with the septum as a ring as cell division proceeds, and eventually collapses into a focus at the center of the septum, whereas it localized to the cell periphery in recently divided cells.

11) Subsection “GpsB localizes to mid-cell and co-constricts with the divisome”, end of second paragraph: The localization of GpsB to the cell periphery in adjacent daughter cells instead of to new septal rings suggests that GpsB localizes later than FtsZ. This would be similar to what was observed for S. pneumoniae (Land et al.). It would have strengthened this section of the manuscript to show whether GpsB localizes later than FtsZ, and this would also test the model in Figure 6.

Agreed. We have now noted (subsection “GpsB dynamically localizes to mid-cell in *S. aureus* and co-constricts with the division septum”, last paragraph) the similarity to *S. pneumoniae* and have cited Land et al.

The two‐color time lapse experiment proved a bit difficult for us due to phototoxicity issues. Nonetheless, to investigate the dependence on FtsZ, we have examined the localization of GpsB in cells that were treated with an FtsZ inhibitor. The data are now presented in a new Figure 4A‐F. As the reviewer predicted, GpsB remained localized at the cell periphery, suggesting a genetic dependence on FtsZ. We thank the reviewer for the suggestion.

12) The high critical concentration for S. aureus FtsZ assembly suggests that this FtsZ needs a stimulatory factor for its assembly in vivo, and the massive increase in FtsZ bundling certainly makes a case for GpsB as a candidate for one of these factors. Localizing FtsZ in GpsB-depleted cells would be one way to validate this idea in vivo and provide some type of correlation between in vivo and in vitro results, for which there is none at present (see main comment above).

Agreed. In a new Figure 4G‐K, we investigate the localization of ZapA‐GFP (proxy for FtsZ) in 1) cells overproducing GpsB, and 2) cells in which GpsB is depleted. In GpsB‐overproducing cells, ZapA‐GFP fails to localize, consistent with a model in which excess GpsB may lead to premature dis‐assembly of FtsZ. In the absence of GpsB (prior to cell lysis), interestingly, ZapA‐GFP localizes to mid‐cell, but the localization is very faint. This is consistent with the in vitro data where we claim that robust assembly (bundling) of FtsZ, but not necessarily subcellular localization, requires GpsB.

13) Subsection “GpsB stimulates GTPase activity of FtsZ”: Do the authors know what the approximate concentration of FtsZ is in S. aureus cells? FtsZ can form aberrant structures when it is highly concentrated, and these structures are also dependent on buffer and pH conditions. The authors have shown that S. aureus FtsZ's GTPase activity is heavily dependent on concentration, probably because its assembly is as well. Therefore, using concentrations of FtsZ that are several-fold higher than what is in cells may give artifactual results.

We have now included a discussion comparing our in vitro data to the physiological concentration of FtsZ and GpsB in *S. aureus* (Discussion, second paragraph). Briefly, FtsZ and GpsB are reported to be present at 4452 and 1659 molecules per cell (Zuhlke et al., 2016; primary data found in http://aureowiki.med.unigreifswald.de/SACOL1199 and http://aureowiki.med.uni‐greifswald.de/SACOL1484), which is equivalent to 7.4 × 10^‐21^ moles of FtsZ. If we consider a *S. aureus* cell with a diameter of 0.8 µm, the cell volume is 2.7 × 10^‐16^L, so the molarity of FtsZ in vivo is 27.6 µM: remarkably close to the 30 µM that was required for our purified FtsZ to show appreciable activity. The intracellular concentration of GpsB is 2.7‐fold less than FtsZ, or 10.2 µM‐ again, remarkably close to the 10 µM we used in our in vitro experiments. Thus, we have used FtsZ and GpsB at concentrations that are nearly identical to their in vivo concentrations.

The reviewer raises an important point about artifactual assembly, which we also considered.

However, the structures observed by EM at 30 μm coincide with an *increase* in the rate of GTP hydrolysis, which also occurs at around 30 μM. We therefore propose that the structures observed are not likely artifactual because they track with increased specific activity. If structures were aberrant and artifactual (i.e., aggregates), one would expect the rate of enzyme activity to decrease.

14)Subsection “GpsB stimulates GTPase activity of FtsZ”: The 3-fold stimulation of GTPase by GpsB is significant and interesting, but of course it is relative to a very low basal GTPase activity that, at a more physiological concentration of 10 μM, calculates to about 3 GTP hydrolyzed per FtsZ per hour. So that means GpsB can bring that number up to 10. The authors should discuss whether those rates of hydrolysis make physiological sense given the doubling time of S. aureus and the likelihood of FtsZ treadmilling at the septal ring (see two recent Science papers on this topic).

1) Please see our detailed response to the physiological concentration of FtsZ and GpsB in point #13. To summarize, 10 µM is *not* the physiological concentration of FtsZ in *S. aureus*, according to newer proteomic data (Zuhlke et al., 2016). The intracellular concentrations of FtsZ and GpsB in *S. aureus* (~28 µM and ~10 µM, respectively) are nearly identical to the concentrations that displayed activity in vitro in our experiments. Our data suggests that FtsZ polymers and regulators appear poised at the threshold between assembly and disassembly, enabling tight control over the division process.

2) Agreed. The two papers that the reviewer referred to were still relatively recent reports at the time of our first submission, so we hesitated in invoking treadmilling explicitly, even though we felt that our data were consistent with such an activity. We have now included an extensive discussion of how we envision our data would be consistent with this activity in vivo. Specifically, we propose that FtsZ polymers would be bundled and organized by GpsB and, following this step, treadmilling would increase as GTP hydrolysis increases. We have also modified our model figure (Figure 7) to incorporate the role of treadmilling in the context of our model.

15) "did not appreciably stimulate" seems to be true, but perhaps the data suggest that GpsB might inhibit GTPase activity?

The mean hydrolysis rate of FtsZ in the presence of GpsB(L35S) is well within the first quartile of data we collected when FtsZ was incubated with WT GpsB (Figure 5F). Combined with the inability of the L35S variant to interact with FtsZ (Figure 6A), we feel uncomfortable suggesting that the GpsB(L35S) negatively influences FtsZ activity and therefore prudently stated that it does not stimulate it. We hope the reviewer agrees with our statement.

16) How reproducible is the separation of GpsB into hexamer and dodecamer fractions as shown in Figure 4C?

We have now indicated in the legend for Figure 5 that the chromatogram is representative of at least three independent purifications.

17) The y axes in Figure 4D-E are not the same as in 4C, where the rate per mol of FtsZ is a more intuitive way to present the data. Perhaps these could be harmonized.

Agreed. The graphs in Figure 5D‐F are now expressed either in pmol/min or pmol/min/pmol, as appropriate.

18) The authors conclude that GpsB stimulates FtsZ GTPase activity sub-stoichiometrically. Does increasing GpsB concentration stimulate FtsZ's GTPase activity further?

We were ourselves interested in this question. However, it was difficult to achieve higher concentrations than 10 µM for GpsB in the reaction so, combined with our observation that FtsZ hydrolysis jumped drastically between 8 and 10 µM GpsB, we left 10 µM as the highest concentration we tested. We hope this will suffice for this initial report.

19) What is the GpsB:FtsZ ratio in cells? How close to this ratio are the experimental conditions?

Please see the detailed response to this question in point #13 above. Briefly, the calculated intracellular concentration of FtsZ is ~28 µM and that of GpsB is ~10 µM‐ nearly identical to the 30 µM FtsZ and 10 µM GpsB that displayed activity in our in vitro experiments.

20) Subsection “GpsB interacts with and bundles FtsZ polymers”, second paragraph and Figure 5: Despite the claim in the text, no obvious structures are visible in the micrographs other than those in F-G.

On the advice of the other reviewers, we have now replaced the fluorescence micrographs with electron microscopy images to demonstrate lateral interactions between FtsZ polymers. The data are now presented as a new Figure 6B‐I.

21) Subsection “GpsB interacts with and bundles FtsZ polymers”, last paragraph and Figure 5N: it looks like FtsZ polymerized at the same rate in the presence of GpsB, just more extensively, so the wording should be changed. And as mentioned elsewhere, "polymerized" likely reflects bundling of protofilaments, not polymerization of single protofilaments.

Agreed. We have now largely replaced “polymerized” with the more generic term “assembly”.

22) How reproducible is the disappearance of fluorescent FtsZ polymers and loss of light scattering signal in response to GpsB? This is certainly an intriguing result, but needs more rigorous corroboration before it can be a major conclusion of the paper (and in the title). Even then, any conclusion about GpsB regulation of FtsZ assembly needs to come with the caveat that GpsB likely interacts with multiple divisome proteins (e.g. see Rued et al., 2017, for examples in S. pneumoniae) that may modulate its activities.

Both experiments were, of course, very reproducible (n>3). The fluorescence data are no longer presented, on the advice of the other reviewers, and has been replaced by EM images to show bundling. To dovetail our results with the treadmilling model of FtsZ, we have now rewritten the manuscript to emphasize that the increase in GTPase activity and the disassembly observed in vitro is likely linked to the increase in treadmilling activity of FtsZ in vivo. As such, the title also deemphasizes the in vitro FtsZ “dis‐assembly” result and now simply states that GpsB regulates FtsZ dynamics.

Regarding the contribution of other divisome proteins, it was certainly not our intention to suggest that no other divisome protein influences FtsZ. We have, though, reported that GpsB *directly* interacts with FtsZ which we have shown in vitro is *sufficient* to modulate FtsZ’s behavior in the absence of any other factors, and the model that we have presented, fitting with the scope of the paper in defining the function of GpsB in *S. aureus*, was meant to be limited to what we report in the paper. Since, in vivo, additional divisome components most certainly contribute to FtsZ’s dynamics, we invoked in the Discussion section the possibility, for example, that EzrA may collaborate with GpsB in *S. aureus* (Discussion, fifth paragraph). Since the interaction of GpsB with the other cell division proteins has not been explicitly reported yet in *S. aureus*, we have not incorporated the contributions of the other factors in our model. We request that the contributions of other proteins may be introduced in models presented in future reports, once we have a better understanding of *S. aureus* cell division.

Finally, we thank the reviewer for pointing out Rued et al. We have now referenced the paper in the Introduction (second paragraph).

23) Discussion, first paragraph: As there is no in vivo evidence to back up the in vitro data, this conclusion seems premature.

We have removed this conclusion from the revised manuscript.

24) Discussion, second paragraph: the logic here is cloudy, as it is not clear why depletion of a protein that both positively and negatively regulates FtsZ assembly could result in incorrect timing of cell division, particularly as this incorrect timing was not directly shown (see comments above).

Agreed. In the revised manuscript, the model has been revised to suggest that bundling by GpsB leads to increased FtsZ GTPase activity in vitro (which, in the GTP‐limiting conditions, eventually leads to disassembly detected by light scatter). in vivo, we suggest that this translates to the activation of FtsZ’s treadmilling activity (i.e., GTP cycling). Consistent with this, depletion of GpsB arrests cell division in vivo.

25) "GpsB rapidly colocalized with the cell division machinery at the onset of cytokinesis". This is partially true, but perhaps more importantly, there seems to be a delay between the arrival of FtsZ and that of GpsB, because GpsB is not localized to the ring (and yet is localized to the membrane) in newborn cells. If GpsB is important for FtsZ ring assembly, which is implied by the model, how might that be reconciled?

Agreed. We have now simply stated that “…GpsB co‐localized with the cell division machinery”, not “rapidly”. In the revised manuscript, we have also shown that localization of ZapA (FtsZ proxy) occurs in the absence of GpsB, but is very faint. Together with the in vitro bundling data, we propose that GpsB permits the “robust” assembly of FtsZ, even though GpsB is not responsible for FtsZ localization at mid‐cell, per se.

26) "consistent with its role in modulating the activity of FtsZ". It is also consistent with its ability to bind a divisome component, which could be FtsZ but also could be something else such as EzrA.

While it is of course possible to invoke other factors, we humbly proposed that the *sufficiency* of GpsB to influence FtsZ’s behavior in vitro leads to the parsimonious model that overproduction of GpsB inhibits FtsZ in vivo. This model is also consistent with our observation that overproduction of GpsB in *B. subtilis* led to filamentation, even in the absence of EzrA. To reiterate, the scope of this paper is, above all, to report that GpsB can directly interact with FtsZ in *S. aureus*. As mentioned above, we have tried to be cautious in applying lessons learned from other (even closely related) organisms, because the function of this protein seems to vary from species to species and, with respect to bundling activity, also behaves differently from other known bundlers. That said, we have not completely ignored the contribution of EzrA to our model and do indeed invoke it in the Discussion section (fifth paragraph).

27) Discussion, third paragraph: In their model, the authors assume that FtsZ disassembly is a feature of ring constriction. While this may be true, there is also published evidence that increased FtsZ filament density is important for ring constriction. At this point, it is totally unclear what happens to FtsZ assembly state during constriction in S. aureus.

Agreed. In the revised manuscript, we have modified our model to incorporate the notion that FtsZ GTPase activity, and hence GTP turnover, is linked to treadmilling in vivo. in vitro, this is reported as disassembly due to GTP depletion. We have revised the manuscript to reflect this in a refined model

(Figure 7).

28) Discussion, third paragraph and following are indeed very speculative considering that they are largely based on one gel filtration experiment.

We agree that this is highly speculative, which is why we decided to place it in the Discussion section. Indeed, we began the sentence with “It is tempting to *speculate*…” so that there would be no doubt that this is not a conclusion and rather a discussion point for further study. If the reviewer wishes, we can certainly remove this entire discussion, but we feel that such speculation in the Discussion section is valuable, especially when we make it clear that we are presenting a testable hypothesis (that is beyond the immediate scope of the current paper).

29) Disussion, third paragraph: Might another model be that the L35S mutant GpsB fails to redistribute in the cell because it interacts poorly with FtsZ?

Definitely! We have now indicated (Discussion, fifth paragraph) that the L35S variant may not interact with FtsZ. We thank the reviewer for the suggestion.

30) Discussion, third paragraph: While the correlation between EzrA and GpsB may be true, it is misleading to correlate them because they both have similar depletion phenotypes; depletion of any protein important for S. aureus cell division will result in cell enlargement.

This is certainly true. As mentioned above, this was an effort to bring into the Discussion section other cell division components, that have been reported in *S. aureus* to be linked to GpsB so as not to present the GpsB/FtsZ interaction in isolation. If the reviewer would like us to remove any mention of EzrA in this capacity in the Discussion section, we can certainly do so and limit the discussion solely to GpsB and FtsZ.

31) Discussion, last paragraph: If this model is true, then an interesting and straightforward experiment would be to determine if there is a time lag in GTP hydrolysis compared with assembly. The prediction is that assembly should occur prior to hydrolysis.

We have revised our model to instead describe that GpsB promotes bundling and increases GTP turnover, likely through increasing local concentrations of FtsZ. Bundling appears to initiate very rapidly by light scatter (Figure 6L), and hence no detectable lag is expected based on the methods described here. We also did not observe a lag by monitoring GTP hydrolysis under the conditions tested (Figure 5C). However, this is an interesting suggestion and has also been reported for FtsZ from *S. pneumoniae,* which exhibits a lag in a time course of GTP hydrolysis whereby peak hydrolysis activity occurs simultaneously with maximal polymerization detected by light scattering amplitude (Salvarelli, et al., 2015). If *Sa* FtsZ behaves similarly, then we would expect to reach maximal hydrolysis at 10‐30 min after addition of nucleotide, the peak by light scattering. Under this scenario, if there were a lag, we would have detected it in a 10 min time course experiment; however, no lag was detected.

Notwithstanding, even a modest increase in concentration (i.e., 33%, which is equivalent to the increase from 30 to 40 μM) more than doubles the rate of GTP hydrolysis, so detectable lag time measurements in the absence and presence of GpsB, which very rapidly induces bundling, would likely require rapid mix/stopped flow experiments, which we hope the reviewer agrees is well beyond the scope of this manuscript.

32) Figure 6: although quite a bit of text early in the paper is devoted to the aberrant nucleoid segregation in response to excess or depleted GpsB levels, the nucleoid is ignored in the model.

We agree. In this preliminary report, the unusual nucleoid segregation was totally distracting and remains a mystery to us. We have therefore left this data out of the current manuscript and will attempt to follow it up at a later date.

33) Figure 6: the model for GpsB is reminiscent of a model for ZipA and FtsA in E. coli, with ZipA as the FtsZ bundler and FtsA as the debundler.

We thank the reviewer for the suggestion. We have now compared our model to that of the combined action of ZipA and FtsA in *E. coli* (Discussion, last paragraph).

34) Do the authors know whether the His_6_ tags on both of the proteins have any aberrant effect on their interactions?

This is an important point. All of the biochemistry experiments in the revised manuscript were replaced with experiments performed with a tag‐less version of FtsZ. The His6 tag on GpsB likely does not impede GpsB function because GpsB‐His6 induces filamentation when overexpressed in *B. subtilis*.

[Editors' note: the author responses to the re-review follow.]

Essential revisions:It is proposed that higher FtsZ bundling and thus higher local concentrations lead to higher GTPase activity. As mentioned in the Discussion this would be in contrast to other proteins such as E. coli ZapA or ZapC that bundle FtsZ without activating GTPase activity. The sedimentation results show nicely that GpsB interacts with FtsZ; but if GpsB bundles FtsZ, why then would its addition not increase the amount of FtsZ in the pellet fraction? Is the fraction of FtsZ bundles already maximized in these conditions? This should be commented upon because the sedimentation results are currently at odds with the light scattering experiments.

We thank the reviewers for the suggestion. The original intention of the sedimentation experiment in Figure 6A was to test the interaction between GpsB and assembled FtsZ polymers. To maximize FtsZ pelleting for this purpose, the experiment was performed with the non-hydrolyzable GMPCPP and at very high relative centrifugation force (130k × g). At this RCF, the experiment as originally performed likely did not distinguish between small polymers, large polymers, and bundles, which is probably why the addition of GpsB did not result in an increase in FtsZ in the pellet fraction.

We have now modified the protocol by centrifuging the reactions at slower speeds. As predicted by the reviewer, we observed differential pelleting behavior of FtsZ in the presence of GpsB (and nucleotide), consistent with bundling. Furthermore, to test the alternative model posed below by the reviewers, we also performed the experiment at lower speed either in the presence of GTP or GMPCPP. The results are now presented as a new Figure 6—figure supplement 1. For +GTP experiments we empirically determined that, when centrifuged at 90k × *g*, addition of GpsB to the FtsZ assembly reaction resulted in increased FtsZ in the pellet fraction, consistent with the reviewers’ prediction that centrifugation should reveal additional FtsZ in the pellet if FtsZ polymers were indeed bundling. For +GMPCPP experiments, the GMPCPP-stabilized FtsZ polymers are much longer, as observed by TEM in the new Figure 6—figure supplement 1D, and we empirically determined that, when centrifuged at 20k × *g*, addition of GpsB again resulted in increased FtsZ in the pellet fraction. The modified centrifugation experiments are now consistent with the light scattering experiments. In sum, we have now demonstrated using three different techniques (differential centrifugation, 90° light scattering, and electron microscopy) that GpsB promotes the assembly of higher order FtsZ structures. Additionally, the new differential centrifugation assay performed with GMPCPP (accompanied by additional electron microscopy studies – please see below for an expanded discussion) indicate that GTP hydrolysis by FtsZ is not required for the GpsB-mediated assembly of higher order FtsZ structures.

Since the original sedimentation experiment showed robust FtsZ pelleting (by using GMPCPP and high RCF), and the reviewers thought it was convincingly demonstrated GpsB interaction with FtsZ, we have retained that experiment in Figure 6A. The new differential centrifugation experiments are now included in a new Figure 6—figure supplement 1.

An alternative model consistent with the sedimentation data that the authors might consider is that GTP hydrolysis is not a consequence of FtsZ bundling but instead is required for bundling, perhaps because GpsB-mediated FtsZ bundling only occurs after FtsZ subunits become poised for turnover. In other words, GpsB stimulates GTP hydrolysis by FtsZ, which then triggers a conformational change that stimulates lateral interactions between FtsZ protofilaments. This could be tested by repeating the sedimentation with GTP instead of GMPCPP, matching the conditions in the light scattering experiment. If enough GTP is provided, e.g. 4 mM or so, the low hydrolysis rate of FtsZ makes it unlikely that GTP will be exhausted during the experiment. The prediction is that the ability of FtsZ to hydrolyze GTP will allow GpsB to bundle FtsZ and consequently pellet much more efficiently. This also predicts that FtsZ bundles would not be detectable by EM in the presence of GpsB and GMPCPP instead of GTP.

The revised sedimentation data (please see the discussion above) are now only consistent with a model in which GTP hydrolysis is not required for FtsZ bundling. To additionally test the alternative model proposed by the reviewers (that increased GTP hydrolysis drives FtsZ bundling), we 1) examined FtsZ assembly in the presence of GpsB and GMPCPP using electron microscopy, as suggested by the reviewers; and 2) performed the differential centrifugation experiment with GMPCPP to monitor the assembly of higher order FtsZ structures.

1) Electron microscopy. In the presence GpsB and GMPCPP, we observed increased lateral association of FtsZ filaments, indicating that GTP hydrolysis is not required for GpsB-mediated bundling of FtsZ. The results are therefore inconsistent with a model in which GTP hydrolysis drives bundling.

2) Differential centrifugation. As noted above, we observed increased FtsZ pelleting even in the presence of GMPCPP, indicating that GTP hydrolysis is not required for the assembly of higher order FtsZ structures, again inconsistent with the alternative model.

We have now experimentally considered the proposed alternative model and conclude that the data are consistent with a model in which bundling precedes GTP hydrolysis. A simple model to explain the data is that GpsB promotes bundling of FtsZ filaments, thereby increasing the local concentration of FtsZ, which then permits GTP hydrolysis. in vitro, GTP hydrolysis results in the depolymerization of FtsZ polymers (and disassembly of bundles) which is observed in the light scattering experiments in Figure 6.